# Inverse Reinforcement Learning in a Continuous State Space with Formal Guarantees

**Gregory Dexter**
Department of Computer Science
Purdue University
West Lafayette, IN, USA
`gdexter@purdue.edu`

**Kevin Bello**
Department of Computer Science
Purdue University
West Lafayette, IN, USA
`kbellome@purdue.edu`

**Jean Honorio**
Department of Computer Science
Purdue University
West Lafayette, IN, USA
`jhonorio@purdue.edu`

## Abstract

Inverse Reinforcement Learning (IRL) is the problem of finding a reward function which describes observed/known expert behavior. The IRL setting is remarkably useful for automated control, in situations where the reward function is difficult to specify manually or as a means to extract agent preference. In this work, we provide a new IRL algorithm for the *continuous* state space setting with *unknown* transition dynamics by modeling the system using a basis of *orthonormal functions*. Moreover, we provide a proof of correctness and formal guarantees on the sample and time complexity of our algorithm. Finally, we present synthetic experiments to corroborate our theoretical guarantees.

## 1 Introduction

Reinforcement learning in the context of a Markov Decision Process (MDP) aims to produce a policy function such that the discounted accumulated reward of an agent following the given policy is optimal. One difficulty in applying this framework to real world problems is that for many tasks of interest, there is no obvious reward function to maximize. For example, a car merging onto a highway must account for many different criteria, e.g., avoiding crashing and maintaining adequate speed, and there is not an obvious formula of these variables producing a reward which reflects expert behavior [3]. The Inverse Reinforcement Learning (IRL) problem consists of having as input an MDP with no reward function along with a policy, and the goal is to find a reward function so that the MDP plus the learned reward function is optimally solved by the given policy. Describing expert behavior in terms of a reward function has been shown to provide a succinct, robust, and interpretable representation of expert behavior for purposes of reinforcement learning [3]. Furthermore, this representation of agent preference can give insight on the preferences or goals of an agent.

Despite much work over the years, there still exist major limitations in our understanding of IRL. One major gap is the lack of algorithms for the setting of continuous state space. This setting is of primary importance since many promising applications, such as autonomous vehicles and robotics, occur in a continuous setting. Existing algorithms for IRL in the continuous setting generally do not directly solve the IRL problem, but instead take a policy matching approach which leads to a less robust description of agent preference [3]. Finally, there is a lack of formal guarantees for IRL methods which make principled understanding of the problem difficult.

35th Conference on Neural Information Processing Systems (NeurIPS 2021).

For a comprehensive list of results on the IRL problem, we refer the reader to [3] as we next describe works that are closest to ours. Throughout the years, various approaches have been proposed to solve the IRL problem in the continuous state setting. Aghasadeghi and Bretl [1] used a finite set of parametric basis functions to model the policy and reward function for a policy matching approach. Boularias et al. [6] matched the observed policy while maximizing Relative Entropy and applied this method to continuous tasks through discretization. Levine and Koltun [12] maximized the likelihood of the observed policy while representing the reward function as a Gaussian Process. Finn et al. [8] modeled the reward function through a neural network to allow for more expressive rewards. Ramponi et al. [18] and Pirotta and Restelli [17] search for an optimal reward based on minimizing the gradient of the continuous policy. Metelli et al. [14] selects a reward function over a set of basis functions such that the policy gradient is zero. These previous methods rely on heuristic arguments and empirical results to support each approach, but do not provide any theoretical guarantees or the insight that comes with them. Finally, Komanduru and Honorio [11] and Metelli et al. [15] provided theoretical analysis similar to our work, but only for the finite state space setting.

**Contributions.** In contrast to the work above, this paper provides a *formally guaranteed solution* to the IRL problem over a continuous state space when transition dynamics of the MDP are *unknown*. We accomplish this by representing the estimated transition dynamics as an infinite sum of basis functions from $\mathbb{R}^2 \to \mathbb{R}$. Then, under natural conditions on the representations, we prove that a linear program can recover a correct reward function. We develop a model of IRL in the continuous setting using the idea of infinite matrices, which in contrast to the more typical non-parametric functional analysis approach, allow us to derive *"explicit, computable, and meaningful error bounds"* [20]. We exhibit our main theorem by proving that if the transition functions of the MDP have appropriately bounded partial derivatives, then by representing the transition functions over a trigonometric basis, our algorithm returns a correct reward function with high probability. However, we emphasize that our results apply under a range of appropriate technical conditions and orthonormal bases. To the best of our knowledge, we present the first algorithm for IRL in the continuous setting with formal guarantees on correctness, sample complexity, and computational complexity. The total sample and computational complexity of our algorithm depends on the series representations of the transition dynamics over a given basis. However, if a given set of transition functions can be represented with sufficiently small error over a given basis by a $k \times k$ matrix of coefficients, then our algorithm (see Algorithm 2) returns a solution to the IRL problem with probability at least $1 - \delta$ in $\mathcal{O}(k^2 \log \frac{k}{\delta})$ samples by solving a linear program with $\mathcal{O}(k)$ variables and constraints. We show how the magnitude of $k$ needed, i.e., the size of the representation, depends on characteristics of a given IRL problem. This sample complexity matches that of the discrete case algorithm of [11]. For more detailed complexity results, see Section 4. We experimentally validate our results by testing our IRL algorithm on a problem with random polynomial transition functions. Finally, we show how our algorithm can be efficiently extended to the $d$-dimensional setting.

## 2   Preliminaries

In this section, we first define the Markov Decision Process (MDP) formalizing the IRL problem. We briefly introduce the Bellman Optimality Equation, which gives necessary and sufficient conditions for a solution to the IRL problem. We then describe the problem of estimating the transition dynamics of the MDP. Finally, we show how to represent the Bellman Optimality Criteria for the given MDP in terms of infinite-matrices and infinite-vectors. For now, we restrict our attention to a one dimensional continuous state space.

**Definition 1** *A Markov Decision Process (MDP) is a 5-tuple $(S, A, \{P_a(s'|s)\}_{a \in A}, \gamma, R)$ where*

- $S = [-1, 1]$ *is an uncountable set of states[1],*

- $A$ *is a finite set of actions ,*

- $\{P_a(s'|s)\}_{a \in A}$ *is a set of probability density functions such that $P_a(s'|s) : S \times S \to \mathbb{R}^+$. $P_a(s'|s)$ represents the transition probability density function from state $s$ to state $s'$ when taking action $a$,*

---

[1]While this paper considers $S = [-1, 1]$, $S$ can easily be generalize to any interval.

- $\gamma \in [0, 1)$ *is the discount factor,*

- $R : S \to \mathbb{R}$ *is the reward function.*

We consider the inverse reinforcement problem. That is, given an MDP without the reward function (MDP $\setminus R$) along with a policy $\pi : S \to A$, our goal is to determine a reward function $R$ such that the policy $\pi$ is optimal for the MDP under the learned reward function $R$.

**Bellman Optimality Criteria.** First we define the value function and Q-function

$$V^\pi(s_0) = \mathbb{E}[R(s_0) + \gamma R(s_1) + \gamma^2 R(s_2) + ...|\pi], \tag{1}$$

$$Q^\pi(s, a) = R(s) + \gamma \mathbb{E}[V^\pi(s')], \quad s' \sim P_a(s'|s). \tag{2}$$

Using these terms, we present the *Bellman Optimality Equation* which gives necessary and sufficient conditions for policy $\pi$ to be optimal for a given MDP [21].

$$\pi(s) \in \arg\max_{a \in A} Q^\pi(s, a), \quad \forall\, s \in S. \tag{3}$$

**Estimation Problem.** We impose the additional task of estimating the transition dynamics of the MDP. While the policy is assumed to be exactly known, the transition probability density functions $\{P_a(s'|s)\}_{a \in A}$ are unknown. Instead, we assume that for an arbitrary state $s$, we can sample the one-step transition $s' \sim P_a(\cdot|s)$, as previously done by Komanduru and Honorio [11].

**Infinite Matrix Representation.** We now introduce a natural assumption on the transition probability density functions, which will make further analysis possible.

Let $\{\phi_n(s)\}_{n \in \mathbb{N}}$ be a countably infinite set of orthonormal basis functions over $S$. That is

$$\int_{-1}^{1} \phi_n^2(s)ds = 1, \text{ and } \int_{-1}^{1} \phi_i(s)\phi_j(s)ds = 0, \ \forall\, i \neq j.$$

Examples of bases that fulfill the above conditions are the Legendre polynomials, the Chebyshev polynomials, and an appropriately scaled trigonometric basis.

We assume that the transition probability density functions $\{P_a(s'|s)\}_{a \in A}$ can be represented over the given basis by,

$$P_a(s'|s) = \sum_{i,j=1}^{\infty} \phi_i(s')\mathcal{Z}_{ij}^{(a)}\phi_j(s), \ \ \mathcal{Z}_{ij}^{(a)} \in \mathbb{R}. \tag{4}$$

In Section 5, we demonstrate sufficient conditions on the transition functions to guarantee that our algorithm is correct when using a trigonometric basis as an example. The following proposition justifies considering only reward functions which are a linear combination of $\{\phi_n(s)\}_{n \in \mathbb{N}}$.

**Proposition 1** *If the transition functions can be represented over $\{\phi_n(s)\}_{n \in \mathbb{N}}$ as in Equation 4 and there is a at least one reward function such that the policy is optimal, then there is at least one reward function $R$ equal to a linear combination of $\{\phi_n(s)\}_{n \in \mathbb{N}}$ such that the Bellman Optimally criterion is fulfilled for the given policy.*

(Missing proofs can be found in the Appendix A.)

This justifies considering only reward functions represented by a series such as,

$$R(s) = \sum_{k=1}^{\infty} \alpha_k \phi_k(s), \quad \alpha_k \in \mathbb{R}. \tag{5}$$

The above conditions are equivalent to requiring that the transition function is a linear operator on the function space spanned by the Schauder basis $\{\phi_n(s)\}$. Proposition 1 guarantees that if there exists a reward which optimally solves the IRL problem, then there is an optimal reward in the closure of the function space spanned by $\{\phi_n(s)\}_{n \in \mathbb{N}}$.

It will be useful to identify the coefficients $\mathcal{Z}_{ij}^{(a)}$ and $\alpha_k$ in Equations 4 and 5 as entries of an infinite-matrix and infinite-vector respectively. This identification allows us to keep all the usual definitions of matrix-matrix multiplication, matrix-vector multiplication, matrix transposition, matrix powers, as well as addition and subtraction.

We define additional simplifying notation. Let the infinite matrix $\mathcal{Z}^{(a)}$ be defined as the matrix of coefficients associated with $P_a(s'|s)$. Let the infinite vector $\alpha$ correspond to coefficients of an arbitrary reward function $R$. We drop the superscript $a$ and write $\mathcal{Z}$ where possible. Let $[^k\mathcal{Z}]$ represent the $k$-truncation of $\mathcal{Z}$. That is, $[^k\mathcal{Z}]_{ij} = \mathcal{Z}_{ij}$ for all $i, j \leq k$, and $[^k\mathcal{Z}]_{ij} = 0$ otherwise. Define the complement of the $k$-truncation of $\mathcal{Z}$ as $[^\infty_k\mathcal{Z}] = \mathcal{Z} - [^k\mathcal{Z}]$. Let $\phi(s)$ be an infinite-vector where the $n$-th entry equals $\phi_n(s)$. Note that in contrast to finite matrices, typical objects, such as matrix products, do not always exist in the infinite matrix case. We add additional conditions on the infinite matrices where necessary to ensure that all needed operations are well-defined. This typically amounts to requiring absolute summability of the reward coefficients $\alpha$ and bounded row absolute-summability of the matrix $\mathcal{Z}^{(a)}$.

We define operations on infinite matrices and vectors to follow typical conventions. Let $\mathcal{A}$ and $\mathcal{B}$ be infinite matrices and $\nu$ be an infinite vector. Then let $\|\mathcal{A}\|_\infty$ denote the induced infinity norm defined as $\|\mathcal{A}\|_\infty = \max_{i \in \mathbb{N}} \sum_{j=1}^\infty |\mathcal{A}_{ij}|$. Let the vector $\ell_1$-norm be defined as $\|\nu\|_1 = \sum_{i=1}^\infty |\nu_i|$ and the $\ell_\infty$-norm defined as $\|\nu\|_\infty = \max_{i \in \mathbb{N}} |\nu_i|$. Let $[\mathcal{A} + \mathcal{B}]_{ij} = \mathcal{A}_{ij} + \mathcal{B}_{ij}$. Let the infinite matrix product be defined as $[\mathcal{A}\mathcal{B}]_{ik} = \sum_{j=1}^\infty \mathcal{A}_{ij}\mathcal{B}_{jk}$. Matrix-vector multiplication is defined as $[\mathcal{A}\nu]_i = \sum_{j=1}^\infty \mathcal{A}_{ij}\nu_j$ and $[\nu^\top\mathcal{A}]_j = \sum_{i=1}^\infty \nu_i\mathcal{A}_{ij}$. Matrix transposition is defined as $[\mathcal{A}^\top]_{ij} = \mathcal{A}_{ji}$. Define matrix power as the repeated product of a matrix, i.e., $\mathcal{A}^r = \mathcal{A}\mathcal{A}...\mathcal{A}$, $r$ times. The outer product of infinite vectors $\nu$ and $\mu$ is denoted as $\nu\mu^\top$ and equals an infinite matrix such that $[\nu\mu^\top]_{ij} = \nu_i\mu_j$. These definitions match those of finite matrices and vectors except that the index set is countably infinite. For more information on infinite matrices, we refer the reader to [7].

Finally, we can give an infinite matrix analogue to the equations of Ng and Russell [16] to characterize the optimal solutions. We restrict our definition to only strictly optimal policies as motivated by Komanduru and Honorio [11]. For simplicity, we also assign the given optimal policy $\pi$ as $\pi(s) \equiv a_1$, $\forall s \in S$, as done in [16] and [11]. Following the same intuition as in the discrete case, we can represent the expectation of a reward function after a sequence of transitions through matrix-matrix and matrix-vector multiplication. Therefore, we can use the following definition to write the difference in Q-functions of the optimal action and any other action under the optimal policy:

$$\mathcal{F}^{(a)} = \sum_{r=0}^\infty \gamma^r (\mathcal{Z}^{(a_1)})^r (\mathcal{Z}^{(a_1)} - \mathcal{Z}^{(a)}). \tag{6}$$

**Necessary and sufficient conditions for policy optimality.** The following Lemma gives necessary and sufficient conditions for policy $\pi$ to be strictly optimal for a given MDP of the form described above. The condition is analogous to the condition in [16]. However, a key difference is that entries of the infinite transition matrices represent coefficients of basis functions and not transition probabilities directly. Therefore, our method of proof is necessarily different.

**Lemma 2** *The given policy $\pi(s)$ is strictly optimal for the* $\mathrm{MDP}(S, A, \{P_a(s'|s)\}_{a \in A}, \gamma, R = \sum_{n=1}^\infty \alpha_n\phi_n(s))$ *if and only if,*

$$\alpha^\top \mathcal{F}^{(a)} \phi(s) > 0, \quad \forall s \in S, \ \forall a \in A \setminus \{a_1\}.$$

Lemma 2 will be used in Section 4 to prove correctness of our main result, Algorithm 2, as it provides a relatively easy to verify sufficient condition on the reward function representation.

## 3  Estimating MDP transition dynamics

As stated in the previous section, we consider the problem where the true transition dynamics of the MDP are unknown. Thus, in this section, we propose an algorithm to estimate the coefficient matrix $\mathcal{Z}^{(a)}$ and derive high-probability error bounds on the estimated matrices $\widehat{\mathcal{F}}^{(a)}$. We denote any estimated matrices with a hat, i.e., $\widehat{\mathcal{Z}}^{(a)}$ is the estimation of $\mathcal{Z}^{(a)}$ and $\widehat{\mathcal{F}}^{(a)}$ is derived from $\widehat{\mathcal{Z}}^{(a)}$.

Since Algorithm 1 is based on random samples, in our next theorem we show that, with probability at least $1 - \delta$, Algorithm 1 returns an estimate of the true matrix $\mathcal{Z}^{(a)}$ with bounded error measured by the induced infinity norm.

**Algorithm 1** `EstimateZ`
___
**Input:** Action $a \in A$, iteration count $n \in \mathbb{N}$, truncation parameter $k \in \mathbb{N}$.

1. Sample $\bar{s} \in \mathbb{R}^n$ such that each $\bar{s}_r$ is independent and $\bar{s}_r \sim \text{Uniform}(-1, 1)$.

2. Sample $s' \in \mathbb{R}^n$ such that each $s'_r$ is independent and $s'_r \sim P_a(\cdot|\bar{s}_r)$.

3. Compute $\widehat{\mathcal{Z}}^{(a)} = \frac{2}{n} \sum_{r=1}^{n} \phi(s'_r)\phi(\bar{s}_r)^\top$.

**Output:** return $[^k \widehat{\mathcal{Z}}^{(a)}]$.
___

**Theorem 3** *Let $\widehat{\mathcal{Z}}^{(a)}$ be the output of Algorithm 1 for inputs $(a, n, k)$, $\varepsilon > 0$, and $\delta \in (0, 1)$. If*

$$\|[^\infty_k \mathcal{Z}]^{(a)}\|_\infty \leq \varepsilon \quad and \quad n \geq \frac{8k^2}{\varepsilon^2} \log \frac{2k^2}{\delta},$$

*then $\|\widehat{\mathcal{Z}}^{(a)} - \mathcal{Z}^{(a)}\|_\infty \leq 2\varepsilon$ with probability at least $1 - \delta$.*

Theorem 3 guarantees an $\varepsilon$-approximation of $\mathcal{Z}^{(a)}$ with probability at least $1 - \delta$ in $n \in \mathcal{O}(\frac{k^2}{\varepsilon^2} \log \frac{k}{\delta})$ samples. The condition that the truncation error is less than $\varepsilon$ can be guaranteed by properties of the *true* MDP, as we will demonstrate in Section 5. Next, we show how the error in estimating $\mathcal{Z}^{(a)}$ propagates to the error in the estimate of infinite matrix $\mathcal{F}^{(a)}$.

**Lemma 4** *If for all $a \in A$, $\|\widehat{\mathcal{Z}}^{(a)} - \mathcal{Z}^{(a)}\|_\infty \leq \varepsilon$ and $\|\mathcal{Z}^{(a)}\|_\infty, \|\widehat{\mathcal{Z}}^{(a)}\|_\infty \leq \Delta < \frac{1}{\gamma}$, then for all $a \in A \setminus \{a_1\}$,*

$$\|\widehat{\mathcal{F}}^{(a)} - \mathcal{F}^{(a)}\|_\infty \leq \frac{2\varepsilon}{(1 - \gamma\Delta)^2}.$$

Lemma 4 matches the intuition that the error of the $\{\mathcal{F}^{(a)}\}$ matrices is larger than that of the $\{\mathcal{Z}^{(a)}\}$ matrices due to the infinite summation (see Equation 6). The propagation of error will be smaller if there is a lower discount factor or the entries of the matrices $\{\mathcal{Z}^{(a)}\}$ are smaller.

## 4    The IRL algorithm

In this section, we present our main algorithm that recovers a strictly optimal reward function with high probability under general conditions on the infinite-matrix representations. Particularly, in Theorem 6 we give the precise assumptions on the series representation of the true MDP to guarantee the linear program in Algorithm 2 recovers an optimal reward function with high probability when using the estimated $\widehat{\mathcal{F}}^{(a)}$ matrices described in Section 3.

**Algorithm 2** `ContinuousIRL`
___
**Input:** Discount factor $\gamma \in (0, 1)$, interval cover parameter $c > 0$, number of samples $n \in \mathbb{N}$, truncation parameter $k \in \mathbb{N}$.

1. Compute $\{\widehat{\mathcal{Z}}^{(a)}\}$ by calling Algorithm 1 with parameters $(a, n, k)$ for each $a \in A$.

2. Compute $\{\widehat{\mathcal{F}}^{(a)}\}$, for each $a \neq a_1$ by using the $\{\widehat{\mathcal{Z}}^{(a)}\}$ in step 1 and Equation 6.

3. Compute a set $\bar{S} \subset [-1, 1]$ of $\lceil 2/c \rceil$ elements such that for all $s \in [-1, 1]$, there exists $\bar{s} \in \bar{S}$ such that $|s - \bar{s}| \leq c$.[^2]

4. Solve the following linear program. Denote this solution by $\widehat{\alpha}$.

$$\min_\alpha \|\alpha\|_1 \text{ s.t. } \alpha^\top \widehat{\mathcal{F}}^{(a)} \phi(\bar{s}) \geq 1, \forall \bar{s} \in \bar{S}, a \in A \setminus \{a_1\}.$$

**Output:** return reward function $\widehat{R}(s) = \sum_{i=1}^{k} \widehat{\alpha}_i \phi_i(s)$.
___

[^2]: Let $\lceil \cdot \rceil$ denote the ceiling function.

We now provide an intuitive explanation of Algorithm 2. For a fixed reward vector $\alpha$, the term $\alpha^\top \mathcal{F} \phi(s)$ can be regarded as a function from $S \to \mathbb{R}$. This function has a series representation over the basis functions $\{\phi_n(s)\}_{n \in \mathbb{N}}$ with coefficients given by the infinite-vector $\mathcal{F}^\top \alpha$. Therefore, the Bellman Optimality criteria can be fulfilled if $\alpha$ is chosen so that the coefficients $\mathcal{F}^\top \alpha$ make $\alpha^\top \mathcal{F} \phi(s)$ positive over $S$. However, bounds for the minimum of a function given its series representation are generally not tight enough and tighter approximations require expensive calculations such as solving a semi-definite program [13]. Instead, we exactly compute $\widehat{\mathcal{F}} \phi(\bar{s})$ for all points in a finite covering set $\bar{S}$. Using Lipschitz continuity of $\alpha^\top \widehat{\mathcal{F}} \phi(s)$, we can lower-bound $\alpha^\top \widehat{\mathcal{F}} \phi(s)$ over $S$. Since $\alpha^\top \widehat{\mathcal{F}} \phi(\bar{s})$ is linear in $\alpha$, linear optimization can be used to ensure the lower bound is positive over $S$.

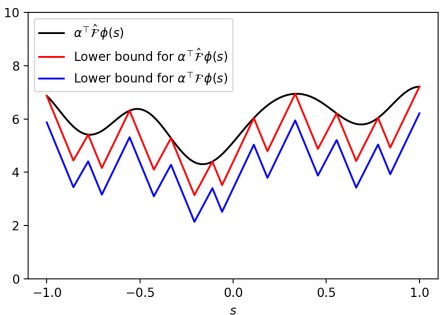

Figure 1: Visualization of Algorithm 2. Here the black line represents $\alpha^\top \widehat{\mathcal{F}} \phi(s)$ and the red line is a lower bound for this function by using the Lipschitz assumption. The blue line is then a lower bound for the true Bellman criteria $\alpha^\top \mathcal{F} \phi(s)$ where the distance between the red and blue scales with $\|\mathcal{F} - \widehat{\mathcal{F}}\|_\infty$. Algorithm 2 finds a sparse $\alpha$ such that $\alpha^\top \mathcal{F}^{(a)} \phi(s)$ is positive over $S$.

We extend the definition of $\beta$-separability given by Komanduru and Honorio [11] to the continuous case in order to provide finite sample guarantees for Algorithm 2. Our experiments indicate that $\beta$ is effective in measuring the intrinsic difficulty of a given IRL problem.

**Definition 2** *($\beta$-separability). We say that the IRL problem is $\beta$-separable if there exists a reward vector $\bar{\alpha}$ such that,*

$$\|\bar{\alpha}\|_1 = 1, \text{ and}$$
$$\bar{\alpha}^\top \mathcal{F}^{(a)} \phi(s) \geq \beta > 0, \ \forall \, s \in S, \ a \in A \setminus \{a_1\}.$$

The condition of $\beta$-separability does not reduce the applicability of the algorithm since every IRL problem with a strictly optimal solution $\bar{\alpha}$ is $\beta$-separable for some $\beta > 0$. Furthermore, $\|\bar{\alpha}\|_1$ is guaranteed to be well-defined, since if there exists $\alpha$ such that $\alpha^\top \mathcal{F}^{(a)} \phi(s) \geq \beta > 0$, then there exists a finite truncation of $\alpha$ called $\bar{\alpha}$ such that $\bar{\alpha}^\top \mathcal{F}^{(a)} \phi(s) \geq \beta' > 0$.

Next, we provide conditions on the smoothness of the expected optimal reward, estimation error of $\mathcal{F}^{(a)}$, and the size of the covering set $\bar{S}$ so that the minimization problem in Algorithm 2 outputs a correct solution.

**Lemma 5** *If the IRL problem is $\beta$-separable, $\|\mathcal{F}^{(a)} \phi'(s)\|_\infty \leq \rho$,[3] $\|\widehat{\mathcal{F}}^{(a)} - \mathcal{F}^{(a)}\|_\infty \leq \frac{\beta - c\rho}{2}$ for all $s \in S$, $a \in A \setminus \{a_1\}$, and $\bar{S} \subset S$ such that there exists $\bar{s} \in \bar{S}$ so $|\bar{s} - s| < c$ for all $s \in S$, then the solution (called $\widehat{\alpha}$) to the optimization problem*

$$\text{minimize}_\alpha \ \|\alpha\|_1$$
$$\text{s. t.} \ \ \alpha^\top \widehat{\mathcal{F}}^{(a)} \phi(\bar{s}) \geq 1$$

*satisfies,*

$$\widehat{\alpha}^\top \mathcal{F} \phi(s) > 0, \ \ \forall \, s \in S.$$

In Figure 1, we provide an intuitive illustration of the correctness of Algorithm 2, which relates to the blue line denoting $\alpha^\top \mathcal{F}^{(a)} \phi(s)$ being positive over $S$ for a given $a \in A$. First, the difference

---

[3]Let $\phi_n'(s) = \frac{d}{ds} \phi_n(s)$.

between the red and blue line must be small, which corresponds to $\|\widehat{\mathcal{F}}^{(a)} - \mathcal{F}^{(a)}\|_\infty$. Secondly, the piece-wise linear bounds composing the red line must be sufficiently "high". The output $\widehat{\alpha}$ will be chosen so that all points on the black curve corresponding to $\bar{s} \in \bar{S}$ will equal at least 1.

We now have the necessary components to prove general conditions on the infinite matrix representation under which Algorithm 2 outputs a solution to the IRL problem with high probability.

**Theorem 6** *If the given MDP without reward function,* $(\mathrm{MDP} \setminus R)$*, is $\beta$-separable, $\beta > c\rho$, and for all $a \in A$, $s \in [-1, 1]$, the following conditions hold:*

$$(i) \ \|\mathcal{F}^{(a)}\phi'(s)\|_\infty < \rho, \quad (ii) \ \|\mathcal{Z}^{(a)}\|_\infty \leq \Delta < \frac{1}{2\gamma}, \quad (iii) \ \|[^\infty_k \mathcal{Z}]^{(a)}\|_\infty \leq \frac{\beta - c\rho}{8}\left(\frac{1}{2} - \gamma\Delta\right)^2,$$

$$n \geq \max\left\{32, \frac{8^3}{(\beta - c\rho)^2(\frac{1}{2} - \gamma\Delta)^4}\right\} k^2 \log\frac{2k^2|A|}{\delta},$$

*then Algorithm 2 called with parameters $(\gamma, c, n, k)$ returns a correct reward function with probability at least $1 - \delta$ by solving a linear program with $k$ variables and $\mathcal{O}(\frac{k|A|}{c})$ constraints.*

**Remark 6.1** *The conditions (i)-(iii) on the infinite-matrix representations of the transition functions given in Theorem 6 are natural for many orthonormal bases. We explain the purpose of each condition and how they relate to assumptions on the MDP without reward function $(\mathrm{MDP} \setminus R)$.*

The first condition (i) enforces that the Bellman Optimality condition $\alpha^\top \mathcal{F}\phi(s)$ is $\rho$-Lipschitz continuous when $\|\alpha\|_1 = 1$. A finite value for $\rho$ exists over the basis $\{\phi_n(s)\}_{n\in\mathbb{N}}$ whenever there is a convergent representation of the derivative of expected optimal reward. Intuitively, this means that expected reward of the optimal policy does not change too quickly with slight changes in the starting state.

The second condition (ii) is needed to enforce that the infinite summation which equals $\mathcal{F}$ (see Equation 6) converges in the induced infinity norm. Conditions on the MDP which enforce this assumption depend on the underlying basis of functions.

Finally, the third condition (iii) is always satisfied by some $k \in \mathbb{N}$ since it is assumed that each transition function has a convergent series representation. Different orthonormal bases provide different guarantees on bounding $k$ given assumptions on the transition function. We will show an example of such bounds for a trigonometric basis in Section 5.

We note that our sample complexity,

$$n \in \mathcal{O}\left(\frac{1}{(\beta - c\rho)^2} k^2 \log\frac{k}{\delta}\right),$$

matches the complexity of the discrete setting algorithm given by Komanduru and Honorio [11] when $\frac{\beta}{c\rho} = \mathcal{O}(1)$.

## 5 Fourier instantiation

The conditions (i) - (iii) of Theorem 6 were chosen to maintain generality while being easily verified by leveraging classic results on orthonormal bases. To illustrate this claim, we prove that Theorem 6 applies to a class of transition functions over a trigonometic basis by using standard methods of proof for Fourier series. Due to space constraints, we leave the technical derivations to Appendix B and focus on the high-level idea in this section. First, we define the basis of trigonometric functions $\{\phi_n(s)\}$ as

$$\phi_n(s) = \cos\left(\lfloor n/2 \rfloor \pi s\right) \quad \text{for all odd } n, \quad \phi_n(s) = \sin\left((n/2)\pi s\right) \quad \text{for all even } n.$$

**Lemma 7** *If $P_a(\tilde{s}|s) = \sum_{i,j=1}^{\infty} \phi_i(\tilde{s}) \mathcal{Z}_{ij}^{(a)} \phi_j(s)$ where $\{\phi_n(s)\}$ are the trigonometric basis functions, $0 < \Delta < \frac{1}{2\gamma}$, $\varepsilon > 0$, and*

$$\left| \frac{\partial^6}{\partial \tilde{s}^3 \partial s^3} P_a(\tilde{s}|s) \right| < \frac{\pi^6 \Delta}{\zeta(3)},$$

*then*

$$(i) \; \|\mathcal{F}\phi'(s)\|_\infty < \frac{4\pi\Delta\zeta(2)}{\zeta(3)}, \quad (ii) \; \|\mathcal{Z}\|_\infty < \Delta, \quad (iii) \; \|[_k^\infty \mathcal{Z}]\|_\infty < \varepsilon,$$

*with truncation parameter $k$ on the order of $k \in \mathcal{O}\left(\sqrt{\frac{\Delta}{\varepsilon}}\right)$, where $\zeta(r) = \sum_{n=1}^{\infty} \frac{1}{n^r}$ is the Riemann zeta function.*

A classical result on Fourier series is that if a periodic function $f$ is of class $\mathbb{C}^p$, then the magnitude of the $n$-th coefficient of its Fourier series is on the order of $\mathcal{O}(1/n^p)$. This can be used to bound the entries of $\mathcal{Z}^{(a)}$ and $\mathcal{F}^{(a)}$. Then, known inequalities of the real Riemann Zeta function and its variants can be used to guarantee the absolute infinite summations that make up conditions (i)-(iii) are appropriately bounded. An additional condition of $\Delta < \frac{1}{4\gamma}$ is then sufficient to guarantee Algorithm 2 has a simplified sample complexity of $\mathcal{O}(\beta^{-3} \log 1/\beta\delta)$ (see Appendix B).

## 6 Experiments

We validate our theoretical results by testing Algorithm 1 and 2 on randomly generated IRL problems using polynomial transition function. We give high level results in this section and leave the details of the experiment in Appendix C.

The following graphs indicate that the theoretical sample complexity of Algorithm 1 given by Theorem 3 is correct. We observe that the number of samples $n$ to achieve estimation error $\varepsilon$ is less than $\frac{8k^2}{\varepsilon^2} \log \frac{2k^2}{\delta}$, the theoretically guaranteed sample complexity.

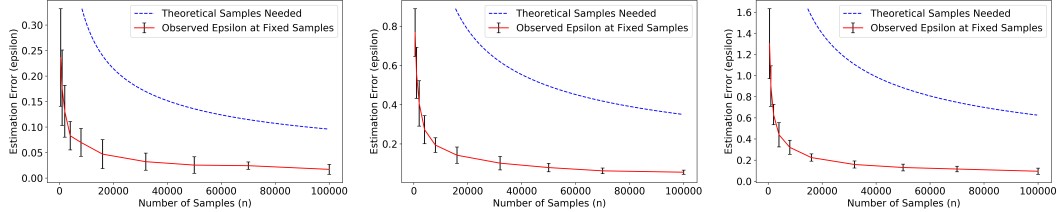

Figure 2: The red line shows the error in estimating the coefficient matrix as $\|[^k\mathcal{Z}] - \widehat{\mathcal{Z}}\|_\infty$ versus the number of samples $n$ using Algorithm 1 for truncation parameter $k = 5$, $k = 15$, $k = 25$. Error bars represent two standard deviations of the observed error. The line 'Theoretical Samples Needed' shows the corresponding sample complexity given by Theorem 3.

Next, we verify the sample complexity of Algorithm 2, while varying the representation size controlled by truncation parameter $k$. We note that the separability parameter $\beta$ has a substantial impact on the sample complexity, as indicated by the complexity result of Theorem 6. We explore this more in Appendix C.

We test the sample complexity result of Algorithm 2 by varying the truncation parameter $k$ and number of samples $n$ for a fixed IRL problem; that is, the actions, discount factor, and transition functions do not change. By keeping the IRL problem fixed, we can derive the following simplified relation from the sample complexity of Theorem 6 where $C > 0$ is a constant.

$$n = \frac{8^3}{(\beta - c\rho)^2(\frac{1}{2} - \gamma\Delta)^4} k^2 \log \frac{2k^2|A|}{\delta}$$

$$\iff \frac{n}{k^2} = \frac{8^3}{(\beta - c\rho)^2(\frac{1}{2} - \gamma\Delta)^4} (\log 2k^2|A| + \log \frac{1}{\delta})$$

$$\iff \log \frac{1}{\delta} = C\frac{n}{k^2} - \log 2k^2|A|$$

The relation $\log \frac{1}{\delta} = C\frac{n}{k^2} - \log 2k^2|A|$ indicates that we expect to see an linear relationship between $\log 1/\delta$ and $n/k^2$. Indeed, we observe this relationship for various values of $k$.

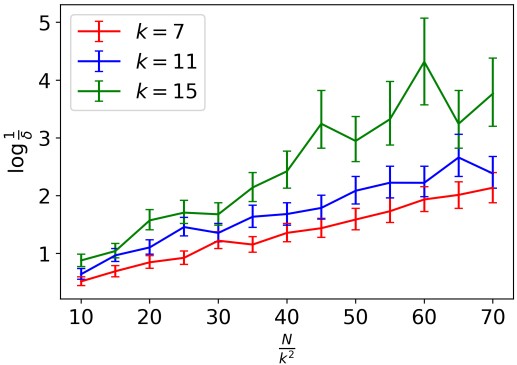

Figure 3: This plot shows the linear relationship between $\log 1/\delta$ and $n/k^2$ while varying the truncation parameter $k$. The error bars show $95\%$ confidence intervals computed via bootstrapping from 240 trials. Other constants: $\beta^{-1} = 26.5$, $\gamma = 0.7$, $c = 0.05$, $\Delta = 1.45$, and $|A| = 3$.

## 7 IRL in $d$-dimensions

In this section, we discuss IRL in the $d$-dimensional setting, i.e., $S = [-1, 1]^d$. One obvious way of extending Algorithm 2 to the $d$-dimensional setting would be to modify Lemma 5 to use $\ell_2$-Lipschitz continuity of $\alpha^\top \mathcal{F}^{(a)}\phi(s)$ with an orthonormal basis $\{\phi_n(s)\}_{n\in\mathbb{N}}$ over $[-1, 1]^d$. Such a $d$-dimensional basis can be constructed by taking the product of 1-dimensional basis functions. This approach has the advantage that it allows for weak assumptions on the $d$-dimensional transition functions, which only need to satisfy the conditions of Theorem 6 when represented over some basis. However, this would require an exponential increase in the size of the covering set $\bar{S}$ used in Algorithm 2 and number of samples needed if the representativeness of the transition functions in each dimension are not restricted further.

It is unlikely that the general $d$-dimensional case be handled without exponential scaling in $d$ since this also occurs in the finite state space setting. To see this, notice that if $|S|$ is finite then an IRL problem with state space $S^d$ can be equivalent to any IRL problem on a space with $|S|^d$ states when the transition function on $S^d$ is arbitrary. Therefore, it is necessary to add additional assumptions to reduce the worst-case computational and statistical complexity.

To simplify the $d$-dimensional setting, we make the assumption that the total $d$-dimensional state space of the MDP can be decomposed as the direct sum of $T$ subspaces of dimension at most $q$ such that the total transition function can be written as a product of transition functions on each of these component subspaces. In other words, we can decompose the state space as follows, $S = S^{(1)} \oplus S^{(2)} \oplus ... \oplus S^{(T)}$ such that $\dim(S^{(j)}) \leq q$. Then, given this decomposition, we require that for all $a \in A$, $P_a(s_1|s_0) = \prod_{j=1}^{T} P_a(s_1^{(j)}|s_0^{(j)})$, where $s^{(j)}$ denotes the projection of $s$ to $S^{(j)}$. If the transition functions of a given MDP satisfy such assumptions, then we can solve the IRL problem in this setting in $\mathcal{O}(T\exp(q))$ times the computational time of solving the 1-dimensional IRL problem due to the following theorem.

**Theorem 8** *Let $(S, A, \{P_a(s_1|s_0)\}, \gamma)$ represent an $(\text{MDP} \setminus R)$ such that $S = [-1, 1]^d$ with some decomposition $S = S^{(1)} \oplus S^{(2)} \oplus ... \oplus S^{(T)}$ such that $S^{(i)} \subset S$, and $\dim(S^{(i)}) \leq q$. Let the transition function operates independently on these components, i.e., $P_a(s_1|s_0) = \prod_{j=1}^{T} P_a(s_1^{(j)}|s_0^{(j)})$ for all $a \in A$ with $s^{(j)} = \text{Proj}_{S^{(j)}}(s)$. If each component IRL problem with transition functions $\{P_a(s_1^{(j)}|s_0^{(j)})\}_{a \in A}$ is solved by reward $R^{(j)}$, then the total $d$-dimensional IRL problem is solved by the reward function $R(s) = \sum_{j=1}^{T} R^{(j)}(s^{(j)})$.*

This assumption is motivated by neurological evidence that human cognition handles the curse-of-dimensionality by decomposing a high-dimensional task into multiple subtasks [9]. This assumption was successfully used by Rothkopf and Ballard [19] in their Modular IRL algorithm to represent human navigation preference, implying that reward functions generated under this assumption could be particularly effective at representing human behavior. Further research on the type of assumptions which are empirically justified and allow strong theoretical guarantees in $d$-dimensions would be an interesting future topic of study.

## 8   Discussion

We have presented a formulation of IRL in the continuous setting by representing the transition functions with infinite-matrices over an orthonormal function basis. We show in Theorem 6 that under natural assumptions on the transition function representations, Algorithm 2 returns a reward function solving the IRL problem with high probability.

Our method of estimating the transition matrices has a sample complexity of $\mathcal{O}\left(\frac{k^2}{(\beta - c\rho)^2} \log \frac{k}{\delta}\right)$, which is same as the discrete IRL method in [11] when $\frac{\beta}{c\rho} = \mathcal{O}(1)$. The computational complexity of Algorithm 2 is dominated by solving a linear program with $k$ variables and $\mathcal{O}(\frac{k|A|}{c})$ constraints. Furthermore, we show how our main theorem, Theorem 6, can be used to guarantee Algorithm 2 for a class of transition functions when represented over a trigonometric basis. We verify our theoretical results by empirically testing our method on randomly generated IRL problems with polynomial transition functions. Finally, we discuss how our method can be applied in the $d$-dimensional setting.

In future work, it would useful to better characterize transition functions that appear in applications of IRL. Much like the wavelet basis allows for sparse representations of audio and video, it is possible that particular orthonormal bases would be better suited for certain tasks where IRL is used. Better understanding of transition functions which appear in practice, such as whether bounds on its partial derviatives are justified, paired with more sophisticated results on orthonormal bases, such as those given by Izumi and Izumi [10], could allow for stronger theoretical guarantees in particular domains. Another potential line of future work could be determining which assumptions allow IRL to scale efficiently with dimension of the state space, while maintaining effectiveness on practical tasks. While our assumption on the decomposability of $d$-dimensional transition functions is supported by prior literature on human motor control, it is unclear how limiting this assumption would be for other applications of IRL. Different methods of simplifying IRL in the continuous $d$-dimensional setting may provide greater flexibility.

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
