# Supplementary material: Inverse Reinforcement Learning in a Continuous State Space with Formal Guarantees

## A  Proofs of lemmas and theorems

### A.1  Additional lemma

**Lemma 9** *Let $s_0$ be the starting state, let $(\boldsymbol{a})_n$ represent a sequence of actions and let $\mathcal{M} = \mathcal{Z}^{(\boldsymbol{a}_r)}\mathcal{Z}^{(\boldsymbol{a}_{r-1})}...\mathcal{Z}^{(\boldsymbol{a}_1)}$ i.e., the product of matrices in $\{\mathcal{Z}^{(a)}\}$ left multiplied in order of the sequence $(\boldsymbol{a})_n$, then,*

$$P(s_r|s_0, (\boldsymbol{a})_n) = \phi(s_r)^\top \mathcal{M}\phi(s_0).$$

**Proof**  Here we use proof by induction. Let $\boldsymbol{X}(s_0, (\boldsymbol{a})_n) = \{s_n|n \in \mathbb{N}\}$ be a random process with a fixed starting point $s_0$ where $(\boldsymbol{a})_n$ is a sequence of actions and $s_r \sim P_{\boldsymbol{a}_r}(s_r|s_{r-1})$. We show that

$$
\begin{aligned}
P(s_2|s_0, (\boldsymbol{a})_n) &= \int_{s_1 \in S} P_{\boldsymbol{a}_2}(s_2|s_1)P_{\boldsymbol{a}_1}(s_1|s_0)ds_1 \\
&= \int_{s_1 \in S} \left( \sum_{k,j=1}^{\infty} \phi_k(s_2)\mathcal{Z}_{kj}^{(\boldsymbol{a}_2)}\phi_j(s_1) \right) \left( \sum_{m,\ell=1}^{\infty} \phi_m(s_1)\mathcal{Z}_{m\ell}^{(\boldsymbol{a}_1)}\phi_\ell(s_0) \right) ds_1 \\
&= \sum_{k,j=1}^{\infty} \phi_k(s_2)\mathcal{Z}_{kj}^{(\boldsymbol{a}_2)} \sum_{m,\ell=1}^{\infty} \mathcal{Z}_{m\ell}^{(\boldsymbol{a}_1)}\phi_\ell(s_0) \int_{s_1 \in S} \phi_j(s_1)\phi_m(s_1)ds_1 \\
&= \phi(s_2)\mathcal{Z}^{(\boldsymbol{a}_2)}\mathcal{Z}^{(\boldsymbol{a}_1)}\phi(s_0),
\end{aligned}
$$

where the last step follows since the integral on line three equals one if $j = m$ and zero otherwise. We note that the interchange of the integral and infinite summation is justified by Section 3.7 in [5], since the coefficients $\mathcal{Z}_{ij}^{(a)}$ are absolutely summable, hence the infinite sum is upper bounded by a constant for all $s_2, s_1, s_0$. If there is an infinite matrix $\mathcal{M}$ such that $P(s_r|s_0, (\boldsymbol{a})_n) = \sum_{m,\ell=1}^{\infty} \phi_m(s_r)\mathcal{M}_{m\ell}\,\phi_\ell(s_0)$, then

$$
\begin{aligned}
P(s_{r+1}|s_0, (\boldsymbol{a})_n) &= \int_{s_r \in S} P_{\boldsymbol{a}_{r+1}}(s_{r+1}|s_r)P(s_r|s_0, (\boldsymbol{a})_n)ds_r \\
&= \int_{s_r \in S} \left( \sum_{k,j=1}^{\infty} \phi_k(s_{r+1})\mathcal{Z}_{kj}^{(\boldsymbol{a}_{r+1})}\phi_j(s_r) \right) \left( \sum_{m,\ell=1}^{\infty} \phi_m(s_r)\mathcal{M}_{m\ell}\phi_\ell(s_0) \right) ds_r \\
&= \sum_{k,j=1}^{\infty} \phi_k(s_{r+1})\mathcal{Z}_{kj}^{(\boldsymbol{a}_{r+1})} \sum_{m,\ell=1}^{\infty} \mathcal{M}_{m\ell}\phi_\ell(s_0) \int_{s_r \in S} \phi_j(s_r)\phi_m(s_r)ds_r \\
&= \phi(s_{r+1})\mathcal{Z}^{(\boldsymbol{a}_{r+1})}\mathcal{M}\phi(s_0).
\end{aligned}
$$

We can then conclude the statement of the lemma by induction.

$\blacksquare$

### A.2  Proof of Proposition 1

**Proof**  By Lemma 9, given a fixed sequence of actions $(\boldsymbol{a})_n$, the $r$-th state $s_r$ under this sequence of actions starting from state $s_0$ has a distribution that can be represented over the basis $\{\phi_n(s)\}$. In other words, there exists an infinite matrix $\mathcal{M}$ such that,

$$P(s_r|s_0, (\boldsymbol{a})_n) = \sum_{i,j=1}^{\infty} \phi_i(s_r)\mathcal{M}_{ij}\phi_j(s_0).$$

We now define the projection $R'(s)$ of reward function $R(s)$ to the span of $\{\phi_n(s)\}$. Let $R'(s) = \sum_{k=1}^{\infty} \alpha_k \phi_k(s)$ such that,

$$\alpha_k = \int_{s \in S} \phi_k(s) R(s) ds$$

For any $r \in \mathbb{N}$, $r > 0$, starting state $s_0$, and action sequence $(\boldsymbol{a})_n$, we can write

$$E[R(s_r)|s_0, (\boldsymbol{a})_n] = \int_{s_r \in S} R(s_r) P(s_r|s_0, (\boldsymbol{a})_n) ds_r$$

$$= \int_{s_r \in S} R(s_r) \left( \sum_{i,j=1}^{\infty} \phi_i(s_r) \mathcal{M}_{ij} \phi_j(s_0) \right) ds_r$$

$$= \sum_{i,j=1}^{\infty} \mathcal{M}_{ij} \phi_j(s_0) \int_{s_r \in S} R(s_r) \phi_i(s_r) ds_r$$

$$= \sum_{i,j=1}^{\infty} \mathcal{M}_{ij} \phi_j(s_0) \alpha_i$$

$$= \sum_{i,j=1}^{\infty} \mathcal{M}_{ij} \phi_j(s_0) \int_{s \in S} \alpha_i \phi_i(s_r) \phi_i(s_r) ds_r$$

$$= \int_{s \in S} \left( \sum_{k=1}^{\infty} \alpha_k \phi_k(s_r) \right) \left( \sum_{i,j=1}^{\infty} \phi_i(s_r) \mathcal{M}_{ij} \phi_j(s_0) \right) ds_r$$

$$= \int_{s_r \in S} R'(s_r) P(s_r|s_0, (\boldsymbol{a})_n) ds_r$$

$$= E[R'(s_r)|s_0, (\boldsymbol{a})_n].$$

Therefore, the expected reward under any sequence of actions for reward $R$ is the same as for the projected reward $R'$ for any state $s_r$ where $r > 0$. The reward at the starting state, $R(s_0)$ does not depend on the policy. Therefore, the value of $R(s_0)$ does not change whether a policy is optimal or not. We can conclude that, regarding optimality of $\pi$, reward function $R$ and $R'$ are equivalent. ∎

### A.3  Proof of Lemma 2

**Proof** For notational simplicity, let $\mathcal{T} \equiv \mathcal{Z}^{(a_1)}$. Recall that the value function is defined as,

$$V^{\pi}(s_0) = R(s_0) + \sum_{r=1}^{\infty} \gamma^r E[R(s_r)|s_0, \pi]$$

As motivated by Proposition 1, we only consider reward function $R$ which is a (potentially infinite) sum of basis functions $\phi_n(s)$. Since in our case $\pi \equiv a_1$, we can apply Lemma 9 with action sequence $(\boldsymbol{a})_n$ such that $\boldsymbol{a}_n = a_1$ for all $n \in \mathbb{N}$, then

$$V^{\pi}(s_0) = R(s_0) + \sum_{r=1}^{\infty} \gamma^r E[R(s_r)|s_0, \pi]$$

$$= R(s_0) + \sum_{r=1}^{\infty} \gamma^r \int_{s_r \in S} R(s_r) P(s_r|s_0) ds_r$$

$$= \alpha^{\top} \phi(s_0) + \sum_{r=1}^{\infty} \gamma^r \alpha^{\top} \mathcal{T}^r \phi(s_0)$$

$$= \sum_{r=0}^{\infty} \gamma^r \alpha^{\top} \mathcal{T}^r \phi(s_0)$$

We use the convention that $[\mathcal{T}^0]_{ij} = 1$ when $i = j$ and $[\mathcal{T}^0]_{ij} = 0$ otherwise. Now, define action sequence $(\boldsymbol{a})_n$ such that $\boldsymbol{a}_1 = a$ and $\boldsymbol{a}_n = a_1$ for all $n > 1$. Let $\boldsymbol{X}(s_0, (\boldsymbol{a})_n) = \{s_n | n \in \mathbb{N}\}$ be a random process with a fixed starting point $s_0$ where $(\boldsymbol{a})_n$ is a sequence of actions and $s_r \sim P_{\boldsymbol{a}_r}(s_r | s_{r-1})$. This expected discounted reward of this random process with parameters $s_0$ and $a$ equals the function $Q^\pi(s_0, a)$, which is equivalent to the following,

$$Q^\pi(s_0, a) = R(s_0) + \gamma E[V^\pi(s_1) | s_0, a]$$

$$= \alpha^\top \phi(s_0) + \gamma \int_{s_1 \in S} V^\pi(s_1) P_a(s_1 | s_0) ds_1$$

$$= \alpha^\top \phi(s_0) + \gamma \left( \sum_{r=0}^\infty \gamma^r \alpha^\top \mathcal{T}^r \right) \mathcal{Z}^{(a)} \phi(s_0)$$

$$= \alpha^\top \phi(s_0) + \sum_{r=0}^\infty \gamma^{r+1} \alpha^\top \mathcal{T}^r \mathcal{Z}^{(a)} \phi(s_0).$$

Therefore, the strict Bellman optimality condition can be written as, for all $a \in A \setminus \{\boldsymbol{a}_1\}$,

$$Q^\pi(s_0, a_1) - Q^\pi(s_0, a) > 0$$

$$\iff \alpha^\top \phi(s_0) + \sum_{r=0}^\infty \gamma^{r+1} \alpha^\top \mathcal{T}^r \mathcal{Z}^{(a_1)} \phi(s_0) - \alpha^\top \phi(s_0) + \sum_{r=0}^\infty \gamma^{r+1} \alpha^\top \mathcal{T}^r \mathcal{Z}^{(a)} \phi(s_0) > 0$$

$$\iff \sum_{r=0}^\infty \gamma^r \alpha^\top \mathcal{T}^r \mathcal{T} \phi(s_0) - \sum_{r=0}^\infty \gamma^r \alpha^\top \mathcal{T}^r \mathcal{Z}^{(a)} \phi(s_0) > 0$$

$$\iff \sum_{r=0}^\infty \gamma^r \alpha^\top \mathcal{T}^r (\mathcal{T} - \mathcal{Z}^{(a)}) \phi(s_0) > 0$$

$$\iff \alpha^\top \left( \sum_{r=0}^\infty \gamma^r \mathcal{T}^r (\mathcal{T} - \mathcal{Z}^{(a)}) \right) \phi(s_0) > 0$$

Note that the last step interchanges the explicit and implicit summations by again using absolute summability of $\alpha$ and $\mathcal{F}^{(a)}$. Substituting the definition of $\mathcal{F}^{(a)}$ into the last line allows us to conclude the following.

$$\alpha^\top \mathcal{F}^{(a)} \phi(s) > 0 \quad \forall\, s \in S,\ a \in A \setminus \{a_1\}.$$

∎

## A.4 Proof of Theorem 3

**Proof** For a fixed action $a$, let $\mathcal{D}$ be a joint distribution of $(s', \bar{s})$ such that $\bar{s}$ is distributed uniformly over $S$ and, given $\bar{s}$, $s' \sim P_a(\cdot | \bar{s})$. In words, we sample the starting state $\bar{s}$ from a uniform distribution and then sample the one-step transition $s'$. Note that,

$$E_\mathcal{D}[\phi_i(s') \phi_j(\bar{s})] = \int_{-1}^1 \int_{-1}^1 P_a(s' | \bar{s}) P_a(\bar{s})\, \phi_i(s') \phi_j(\bar{s}) ds' d\bar{s}$$

$$= \frac{1}{2} \int_{-1}^1 \int_{-1}^1 P_a(s' | \bar{s})\, \phi_i(s') \phi_j(\bar{s}) ds' d\bar{s}$$

$$= \frac{1}{2} \int_{-1}^1 \int_{-1}^1 \left( \sum_{p,q=1}^\infty \phi_p(s') \mathcal{Z}_{pq}^{(a)} \phi_q(\bar{s}) \right) \phi_i(s) \phi_j(\bar{s}) ds' d\bar{s}$$

$$= \frac{1}{2} \int_{-1}^1 \int_{-1}^1 \mathcal{Z}_{ij}^{(a)}\, \phi_i(s')^2 \phi_j(\bar{s})^2 ds' d\bar{s}$$

$$= \frac{1}{2} \mathcal{Z}_{ij}^{(a)}.$$

Therefore, $\widehat{\mathcal{Z}}^{(a)} = \frac{2}{n}\sum_{r=1}^{n}\phi(s_r')\phi(\bar{s}_r)^{\top}$ computed by Algorithm 1 is an unbiased estimator of $\mathcal{Z}^{(a)}$ thus $[^k\widehat{\mathcal{Z}}^{(a)}]$ is an unbiased estimator of $[^k\mathcal{Z}^{(a)}]$. We drop the superscript $(a)$ in the rest of the proof, since the same argument applies for all $a \in A$.

We can then use Hoeffding's inequality to get concentration of measure for each entry in the truncated matrix. Since $\phi_i(s')\phi_j(\bar{s}) \in [-1, 1]$, we have

$$P\left(\left|\frac{2}{n}\sum_{r=1}^{n}\phi_i(s_r')\phi_j(\bar{s}_r) - [^k\mathcal{Z}]_{ij}\right| \geq \varepsilon\right) \leq 2e^{\frac{-n\varepsilon^2}{8}}.$$

Then we can use subadditivity of measure to bound the maximum difference across all entries of $[^k\mathcal{Z}]$.

$$P\left(\max_{i,j\in\mathbb{N}}|\frac{2}{n}\sum_{r=1}^{n}[\phi(s_r')\phi(\bar{s}_r)]_{ij} - [^k\mathcal{Z}]_{ij}| \geq \varepsilon\right) \leq 2k^2 e^{\frac{-n\varepsilon^2}{8}}.$$

Lastly, since we want a bound on the error of the induced infinity norm and not element wise error, we use the following

$$\|[^k\mathcal{Z}]\|_{\infty} \leq k\max_{i,j\in\mathbb{N}}\left|[^k\mathcal{Z}]_{ij}\right|.$$

Therefore, the induced infinity norm error of $\widehat{\mathcal{Z}}$ is less than $\varepsilon$ if the element wise error is less than $\varepsilon/k$. That is,

$$P\left(\left\|\frac{2}{n}\sum_{r=1}^{n}\phi(s_r')\phi(\bar{s}_r) - [^k\mathcal{Z}]\right\|_{\infty} \geq \varepsilon\right) \leq P\left(\max_{i,j\in\mathbb{N}}\left|\frac{2}{n}\sum_{r=1}^{n}[\phi(s_r')\phi(\bar{s}_r)]_{ij} - [^k\mathcal{Z}]_{ij}\right| \geq \frac{\varepsilon}{k}\right)$$

$$\leq 2k^2 e^{\frac{-n\varepsilon^2}{8k^2}}.$$

Now we find the sample complexity for fixed $\delta$ and $\varepsilon$. This is equivalent to $\delta \geq 2k^2 e^{\frac{-n\varepsilon^2}{8k^2}}$, which is also equivalent to $n \geq \frac{8k^2}{\varepsilon^2}\log(\frac{2k^2}{\delta})$.

Then, we can bound $\|\widehat{\mathcal{Z}} - \mathcal{Z}\|_{\infty}$ by the triangle inequality with high probability,

$$\|\widehat{\mathcal{Z}} - \mathcal{Z}\|_{\infty} = \|\widehat{\mathcal{Z}} - ([^\infty_k\mathcal{Z}] + [^k\mathcal{Z}])\|_{\infty}$$
$$\leq \|[^\infty_k\mathcal{Z}]\|_{\infty} + \|\widehat{\mathcal{Z}} - [^k\mathcal{Z}]\|_{\infty}$$
$$\leq 2\varepsilon,$$

which concludes our proof. ∎

## A.5  Proof of Lemma 4

**Proof** For notational simplicity, we drop the superscript $(a)$ and denote $\mathcal{T} \equiv \mathcal{Z}^{(a_1)}$. Recall that,

$$\mathcal{F} = \sum_{r=0}^{\infty}\gamma^r\mathcal{T}^r(\mathcal{T} - \mathcal{Z}).$$

First we bound the error of $\widehat{\mathcal{T}}^{r+1}$ as follows.

$$\|\widehat{\mathcal{T}}^{r+1} - \mathcal{T}^{r+1}\|_{\infty}$$
$$= \|\widehat{\mathcal{T}}^r\widehat{\mathcal{T}} - \mathcal{T}^r\mathcal{T}\|_{\infty}$$
$$= \|\widehat{\mathcal{T}}^r\widehat{\mathcal{T}} - \widehat{\mathcal{T}}^r\mathcal{T} + \widehat{\mathcal{T}}^r\mathcal{T} - \mathcal{T}^r\mathcal{T}\|_{\infty}$$
$$\leq \|\widehat{\mathcal{T}}^r\widehat{\mathcal{T}} - \widehat{\mathcal{T}}^r\mathcal{T}\|_{\infty} + \|\widehat{\mathcal{T}}^r\mathcal{T} - \mathcal{T}^r\mathcal{T}\|_{\infty}$$
$$\leq \|\widehat{\mathcal{T}}^r\|_{\infty}\|\widehat{\mathcal{T}} - \mathcal{T}\|_{\infty} + \|\mathcal{T}\|_{\infty}\|\widehat{\mathcal{T}}^r - \mathcal{T}^r\|_{\infty}$$
$$\leq \|\widehat{\mathcal{T}}\|_{\infty}^r\|\widehat{\mathcal{T}} - \mathcal{T}\|_{\infty} + \|\mathcal{T}\|_{\infty}\|\widehat{\mathcal{T}}^r - \mathcal{T}^r\|_{\infty}$$
$$\leq \Delta^r\varepsilon + \Delta\|\widehat{\mathcal{T}}^r - \mathcal{T}^r\|_{\infty}.$$

Define the recurrence relation $\boldsymbol{T}(r+1) = \Delta^r \varepsilon + \Delta \boldsymbol{T}(r)$ and $\boldsymbol{T}(1) = \varepsilon$. Solving this recursive formula gives the following bound,

$$\|\widehat{\mathcal{T}}^{r+1} - \mathcal{T}^{r+1}\|_\infty \leq \boldsymbol{T}(r+1) \leq (r+1)\varepsilon\Delta^r.$$

We can use the inequality above to prove the bound on the induced infinity norm error of $\widehat{\mathcal{F}}$ as follows.

$$\|\widehat{\mathcal{F}} - \mathcal{F}\|_\infty = \Big\| \sum_{r=0}^{\infty} \gamma^r [\widehat{\mathcal{T}}^r(\widehat{\mathcal{T}} - \widehat{\mathcal{Z}}) - \mathcal{T}^r(\mathcal{T} - \mathcal{Z})] \Big\|_\infty$$

$$= \sum_{r=0}^{\infty} \gamma^r \|\widehat{\mathcal{T}}^{r+1} - \widehat{\mathcal{T}}^r \widehat{\mathcal{Z}} - \mathcal{T}^{r+1} + \mathcal{T}^r \mathcal{Z}\|_\infty$$

$$\leq \sum_{r=0}^{\infty} \gamma^r (\|\widehat{\mathcal{T}}^{r+1} - \mathcal{T}^{r+1}\|_\infty + \|\widehat{\mathcal{T}}^r \widehat{\mathcal{Z}} - \mathcal{T}^r \mathcal{Z}\|_\infty)$$

$$\leq \sum_{r=0}^{\infty} \gamma^r (\varepsilon(r+1)\Delta^r + \|\widehat{\mathcal{T}}^r\|_\infty \|\widehat{\mathcal{Z}} - \mathcal{Z}\|_\infty$$

$$+ \|\widehat{\mathcal{T}}^r - \mathcal{T}^r\|_\infty \|\mathcal{Z}\|_\infty)$$

$$\leq \sum_{r=0}^{\infty} \gamma^r (\varepsilon(r+1)\Delta^r + \Delta^r \varepsilon + r\Delta^r \varepsilon)$$

$$\leq \varepsilon \left( \sum_{r=0}^{\infty} 2\gamma^r (r+1)\Delta^r \right)$$

$$\leq 2\varepsilon \left( \frac{1}{1 - \gamma\Delta} + \frac{\gamma\Delta}{(1 - \gamma\Delta)^2} \right)$$

$$\leq 2\varepsilon \frac{1}{(1 - \gamma\Delta)^2}.$$

Where the second to last step follows from applying the standard formula for solving arithmetico-geometric series. ∎

## A.6   Proof of Lemma 5

**Proof** The basis functions $\{\phi_n(s)\}$ are assumed to be continuously differentiable; the trigonometric basis is an example of such basis. Therefore, $\widehat{\alpha}^\top \mathcal{F}\phi(s)$ is $\rho$-Lipschitz if the absolute value of its derivative is bounded by $\rho$, i.e.

$$|\widehat{\alpha}^\top \mathcal{F}\phi'(s)| \leq \rho, \quad \forall\, s \in S.$$

Since there exists $\bar{s} \in \bar{S}$ for all $s \in S$ such that $|s - \bar{s}| < c$, if $\widehat{\alpha}^\top \mathcal{F}\phi(s)$ is $\rho$-Lipschitz and $\widehat{\alpha}^\top \mathcal{F}\phi(\bar{s}) \geq c\rho$ then,

$$\widehat{\alpha}^\top \mathcal{F}\phi(\bar{s}) \geq c\rho, \quad \forall\, \bar{s} \in \bar{S},$$

implies that for every $s \in S$ there exists an $\bar{s} \in \bar{S}$ such that,

$$\widehat{\alpha}^\top \mathcal{F}\phi(s) \geq \widehat{\alpha}^\top \mathcal{F}\phi(\bar{s}) - c\rho,$$
$$\Rightarrow \widehat{\alpha}^\top \mathcal{F}\phi(s) \geq 0, \quad \forall s \in S.$$

We now bound $\|\widehat{\alpha}\|_1$ in order to assess the maximum effect the error in estimating $\mathcal{F}$ can have on the Bellman Optimality Criteria $\widehat{\alpha}^\top \mathcal{F}\phi(s)$. Since the IRL problem is $\beta$-separable, there exists $\bar{\alpha}$ such that $\|\bar{\alpha}\|_1 = 1$ and $\bar{\alpha}^\top \mathcal{F}\phi(s) \geq \beta$ for all $s \in S$.

Let $G > 0$ and $\tilde{\alpha} = G\bar{\alpha}$. Then $\|\tilde{\alpha}\|_1 = G$ and

$$
\begin{aligned}
\tilde{\alpha}^\top \widehat{\mathcal{F}}\phi(\bar{s}) &= \tilde{\alpha}^\top (\widehat{\mathcal{F}} - \mathcal{F} + \mathcal{F})\phi(\bar{s}) \\
&= \tilde{\alpha}^\top \mathcal{F}\phi(\bar{s}) + \tilde{\alpha}^\top (\widehat{\mathcal{F}} - \mathcal{F})\phi(\bar{s}) \\
&\geq \tilde{\alpha}^\top \mathcal{F}\phi(\bar{s}) - \|\tilde{\alpha}^\top\|_1 \|(\widehat{\mathcal{F}} - \mathcal{F})\phi(\bar{s})\|_\infty \\
&\geq G(\beta - \varepsilon).
\end{aligned}
$$

Therefore, if $G = 1/(\beta - \varepsilon)$ then $\tilde{\alpha}^\top \widehat{\mathcal{F}}\phi(\bar{s}) \geq 1$ and thus $\|\widehat{\alpha}\|_1 \leq 1/(\beta - \varepsilon)$. Therefore,

$$
\begin{aligned}
\widehat{\alpha}^\top \mathcal{F}\phi(\bar{s}) &= \widehat{\alpha}^\top (\mathcal{F} - \widehat{\mathcal{F}})\phi(\bar{s}) + \widehat{\alpha}^\top \widehat{\mathcal{F}}\phi(\bar{s}) \\
&\geq \widehat{\alpha}^\top \widehat{\mathcal{F}}\phi(\bar{s}) - \|\widehat{\alpha}\|_1 \|(\mathcal{F} - \widehat{\mathcal{F}})\phi(\bar{s})\|_\infty \\
&\geq 1 - \frac{\varepsilon}{\beta - \varepsilon}.
\end{aligned}
$$

We can upper bound $|\widehat{\alpha}^\top \mathcal{F}\phi'(s)|$ as follows,

$$
|\widehat{\alpha}^\top \mathcal{F}\phi'(s)| \leq \|\widehat{\alpha}\|_1 \|\mathcal{F}\phi'(s)\|_\infty \leq \frac{\rho}{\beta - \varepsilon}.
$$

Then, we can guarantee that $\widehat{\alpha}^\top \mathcal{F}\phi(s) > 0$ for all $s \in S$ if,

$$
1 - \frac{\varepsilon}{\beta - \varepsilon} > \frac{c\rho}{\beta - \varepsilon} \iff \varepsilon < \frac{\beta - c\rho}{2}.
$$

$\blacksquare$

### A.7 Proof of Theorem 6

**Proof** By Theorem 3, condition $(iii)$ of Theorem 6 and the condition on $n$, calling Algorithm 1 with parameters $(a, n, k)$ returns an estimate $\widehat{\mathcal{Z}}^{(a)}$ such that with probability at least $1 - \frac{\delta}{|A|}$,

$$
\|\mathcal{Z}^{(a)} - \widehat{\mathcal{Z}}^{(a)}\|_\infty \leq \frac{\beta - c\rho}{4}\left(\frac{1}{2} - \gamma\Delta\right)^2.
$$

Applying the union bound guarantees that the above inequality holds for all $a \in A$ with probability at least $1 - \delta$.

Note that $\|\widehat{\mathcal{Z}}^{(a)}\| \leq \Delta + \frac{1}{2\gamma}$ is satisfied for all $a \in A$ since $n \geq 32k^2 \log \frac{2k^2|A|}{\delta}$. Then by Lemma 4,

$$
\begin{aligned}
\|\widehat{\mathcal{F}}^{(a)} - \mathcal{F}^{(a)}\|_\infty &< \frac{2(\beta - c\rho)}{4} \frac{(\frac{1}{2} - \gamma\Delta)^2}{(1 - \gamma(\Delta + \frac{1}{2\gamma}))^2} \\
&< \frac{\beta - c\rho}{2}, \quad \forall\, a \in A \setminus \{a_1\}.
\end{aligned}
$$

The minimization problem at Step 4 of Algorithm 2 fulfills the assumptions of Lemma 5, and thus it returns $\widehat{\alpha}$ such that,

$$
\widehat{\alpha}^\top \mathcal{F}^{(a)}\phi(s) > 0, \quad \forall\, s \in S,\ a \in A \setminus \{a_1\}.
$$

Therefore, Algorithm 2 will return a reward function such that $a_1$ is an optimal policy for the MDP with probability at least $1 - \delta$.

The time complexity of Algorithm 2 is dominated by solving the linear program. Since $\widehat{\mathcal{F}}$ has all zeros beyond the $k$-th column and row, each infinite-matrix $\widehat{\mathcal{F}}$ can be treated as a $k \times k$ matrix. Therefore, the constraints given by

$$
\alpha^\top \widehat{\mathcal{F}}^{(a)}\phi(\bar{s}) \geq 1, \quad \forall \bar{s} \in \bar{S},\ \ a \in A \setminus \{a_1\}.
$$

correspond to $|\bar{S}|(|A| - 1)$ constraints in $k$ variables. Since $|\bar{S}| = \lceil 2/c \rceil$, there are $\mathcal{O}(\frac{k|A|}{c})$ constraints in total. $\blacksquare$

## A.8 Proof of Theorem 8

**Proof**

By the conditions of the Theorem, we have

$$P_a(s_1|s_0) = \prod_{j=1}^{T} P_a(s_1^{(j)}|s_0^{(j)}).$$

We use proof by induction to show that $P_a(s_r|s_0) = \prod_{j=1}^{T} P_a(s_r^{(j)}|s_0^{(j)})$ for all $r \in \mathbb{N}$. First, we prove the inductive step.

$$
\begin{aligned}
P(s_{r+1}|s_0) &= \int_{s_r \in S} P(s_{r+1}|s_r)P(s_r|s_0)ds_r \\
&= \int_{s_r^{(1)}...s_r^{(T)}} \left( \prod_{j=1}^{T} P(s_{r+1}^{(j)}|s_r^{(j)}) \right) \left( \prod_{j=1}^{T} P(s_r^{(j)}|s_0^{(j)}) \right) ds_r^{(1)}...ds_r^{(T)} \\
&= \prod_{j=1}^{T} \int_{s_r^{(j)}} P(s_{r+1}^{(j)}|s_r^{(j)})P(s_r^{(j)}|s_0^{(j)})ds_1^{(j)} \\
&= \prod_{j=1}^{T} P(s_{r+1}^{(j)}|s_0^{(j)}).
\end{aligned}
$$

The base case of $r = 1$ is guaranteed by the conditions of the Theorem. Therefore, we conclude that $P_a(s_r|s_0) = \prod_{j=1}^{T} P_a(s_r^{(j)}|s_0^{(j)})$ for all $r \in \mathbb{N}$.

Next, we use the previous decomposition of $P_a(s_r|s_0)$ to rewrite the Bellman Optimality condition. Recall that $R$ is strictly Bellman-optimal if for all $a \in A \setminus \{a_1\}$ and $s_0 \in S$,

$$E[V^\pi(s_1)|s_0, a_1] > E[V^\pi(s_1)|s_0, a].$$

We can rearrange each side of the above inequality by the following.

$$
\begin{aligned}
E[V^\pi(s_1)|s_0, a] &= \sum_{r=1}^{\infty} \gamma^r \int_{s_r \in S} R(s_r)P_{a_1}(s_r|s_1)P_a(s_1|s_0)Ts_r \\
&= \sum_{r=1}^{\infty} \gamma^r \int_{s_r^{(1)}...s_r^{(T)}} \left( \sum_{i=1}^{T} R^{(i)}(s^{(i)}) \right) \left( \prod_{j=1}^{T} P_{a_1}(s_r^{(j)}|s_1^{(j)})P_a(s_1^{(j)}|s_0^{(j)}) \right) ds_r^{(1)}...ds_r^{(T)} \\
&= \sum_{r=1}^{\infty} \gamma^r \int_{s_r^{(1)}...s_r^{(T)}} \sum_{i=1}^{T} \left( R^{(i)}(s^{(i)}) \prod_{j=1}^{T} P_{a_1}(s_r^{(j)}|s_1^{(j)})P_a(s_1^{(j)}|s_0^{(j)}) \right) ds_r^{(1)}...ds_r^{(T)} \\
&= \sum_{r=1}^{\infty} \gamma^r \sum_{i=1}^{T} \int_{s_r^{(1)}...s_r^{(T)}} R^{(i)}(s^{(i)}) \prod_{j=1}^{T} P_{a_1}(s_r^{(j)}|s_1^{(j)})P_a(s_1^{(j)}|s_0^{(j)})ds_r^{(1)}...ds_r^{(T)} \\
&= \sum_{r=1}^{\infty} \gamma^r \sum_{i=1}^{T} \int_{s_r^{(i)}} R^{(i)}(s^{(i)})P_{a_1}(s_r^{(i)}|s_1^{(i)})P_a(s_1^{(i)}|s_0^{(i)})ds_r^{(i)} \\
&= \sum_{i=1}^{T} \sum_{r=1}^{\infty} \gamma^r \int_{s_r^{(i)}} R^{(i)}(s^{(i)})P_{a_1}(s_r^{(i)}|s_1^{(i)})P_a(s_1^{(i)}|s_0^{(i)})ds_r^{(i)}.
\end{aligned}
$$

The second-to-last step above follows since the integral of any probability distribution equals 1.

Since each 1-dimensional IRL problem is solved by $R^{(j)}(s^{(j)})$,

$$\sum_{r=1}^{\infty} \gamma^r \int_{s_r^{(j)}} R^{(j)}(s_r^{(j)}) P_{a_1}(s_r^{(j)}|s_1^{(j)}) P_{a_1}(s_1^{(j)}|s_0^{(j)}) ds_r^{(j)}$$
$$> \sum_{r=1}^{\infty} \gamma^r \int_{s_r^{(j)}} R(s_r^{(j)}) P_{a_1}(s_r^{(j)}|s_1^{(j)}) P_a(s_1^{(j)}|s_0^{(j)}) ds_r^{(j)}.$$

This implies that,

$$\sum_{i=1}^{T} \sum_{r=1}^{\infty} \gamma^r \int_{s_r^{(j)}} R^{(j)}(s_r^{(j)}) P_{a_1}(s_r^{(j)}|s_1^{(j)}) P_{a_1}(s_1^{(j)}|s_0^{(j)}) ds_r^{(j)}$$
$$> \sum_{i=1}^{T} \sum_{r=1}^{\infty} \gamma^r \int_{s_r^{(j)}} R(s_r^{(j)}) P_{a_1}(s_r^{(j)}|s_1^{(j)}) P_a(s_1^{(j)}|s_0^{(j)}) ds_r^{(j)}$$
$$\Rightarrow$$
$$E[V^\pi(s_1)|s_0, a_1] > E[V^\pi(s_1)|s_0, a].$$

Therefore, reward function $R(s)$ is strictly Bellman optimal for the total IRL problem.

∎

## B    Fourier instantiation

We prove Lemma 7 by splitting the derivations into two lemmas. First we state stronger conditions on the infinite-matrix $\mathcal{Z}$ which are sufficient to satisfy the representation assumptions of Theorem 6. These stronger conditions will be easier to verify using known proof techniques for Fourier series.

**Lemma 10** *Let* $0 < \Delta < \frac{1}{2\gamma}$. *If,*

$$|\mathcal{Z}_{ij}| < \frac{\Delta}{\zeta(3)i^3 j^3}, \quad and \quad |\phi'_n(s)| \leq Cn, \ \ C > 0,$$

*then*

$$\|\mathcal{F}\phi'(s)\|_\infty < \frac{4C\Delta\zeta(2)}{\zeta(3)}, \quad \|[_k^\infty \mathcal{Z}]\|_\infty < \varepsilon, \quad \|\mathcal{Z}\|_\infty < \Delta,$$

*with truncation parameter* $k$ *on the order of* $k \in \mathcal{O}\left(\sqrt{\frac{\Delta}{\varepsilon}}\right)$, *where* $\zeta(r) = \sum_{n=1}^{\infty} \frac{1}{n^r}$ *is the Riemann zeta function and* $\varepsilon > 0$.

**Proof** By the condition of the lemma, $|\mathcal{Z}_{ij}| < \frac{\Delta}{\zeta(3)i^3 j^3}$. First we bound, $\|\mathcal{Z}\|_\infty$,

$$\|\mathcal{Z}\|_\infty = \max_{i \in \mathbb{N}} \sum_{j=1}^{\infty} |\mathcal{Z}_{ij}|$$
$$< \max_{i \in \mathbb{N}} \sum_{j=1}^{\infty} \frac{\Delta}{\zeta(3)i^3 j^3}$$
$$= \frac{\Delta}{\zeta(3)} \sum_{j=1}^{\infty} \frac{1}{j^3}$$
$$= \frac{\zeta(3)\Delta}{\zeta(3)}$$
$$= \Delta.$$

Therefore, we can guarantee that $\|\mathcal{Z}\|_\infty < \Delta$. Next we bound $\|[_k^\infty\mathcal{Z}]\|_\infty$. First, we define the Hurwitz zeta function, $H_s(x)$.

$$H_s(x) = \sum_{n=0}^\infty \frac{1}{(n+x)^s} \qquad (7)$$

Recall that $[_k^\infty\mathcal{Z}]_{ij} = 0$ for all $i,j \le k$. Therefore,

$$\begin{aligned}
\|[_k^\infty\mathcal{Z}]\|_\infty &= \max\left\{ \max_{i\in\{1,\dots,k\}} \sum_{j=k+1}^\infty |\mathcal{Z}_{i,j}|,\ \max_{i\in\{k+1,k+2,\dots\}} \sum_{j=1}^\infty |\mathcal{Z}_{i,j}| \right\} \\
&< \max\left\{ \sum_{j=k+1}^\infty \frac{\Delta}{j^3\zeta(3)},\ \sum_{j=1}^\infty \frac{\Delta}{(k+1)^3 j^3\zeta(3)} \right\} \\
&= \max\left\{ \frac{\Delta}{\zeta(3)} \sum_{j=k+1}^\infty \frac{1}{j^3},\ \frac{\Delta}{(k+1)^3\zeta(3)} \sum_{j=1}^\infty \frac{1}{j^3} \right\} \\
&= \max\left\{ \frac{\Delta}{\zeta(3)} H_3(k+1),\ \frac{\Delta}{(k+1)^3} \right\}
\end{aligned}$$

We can bound the Hurwitz zeta function by combining the following two inequalities where $\psi(x)$ represents the digamma function. The first inequality comes from Theorem 3.1 in [4] and the second comes from Equation 2.2 in [2].

$$H_{s+1}(x) < \frac{1}{s}\exp(-s\psi(x))$$
$$\log x - \frac{1}{x} \le \psi(x) \le \log x - \frac{1}{2x}$$

This gives

$$\begin{aligned}
H_3(k+1) &\le \frac{1}{2}\exp\left( -2\left( \log(k+1) - \frac{1}{k+1} \right) \right) \\
&= \frac{1}{2} e^{-2\log(k+1)} e^{\frac{2}{k+1}} \\
&= \frac{1}{2(k+1)^2} e^{\frac{2}{(k+1)}}.
\end{aligned}$$

We can then bound the truncation error as,

$$\begin{aligned}
\|[_k^\infty\mathcal{Z}]\|_\infty &< \max\left\{ \frac{\Delta}{\zeta(3)} H_3(k+1),\ \frac{\Delta}{(k+1)^3} \right\} \\
&\le \max\left\{ \frac{\Delta e}{2\zeta(3)(k+1)^2},\ \frac{\Delta}{(k+1)^3} \right\}.
\end{aligned}$$

Therefore, $k \in \mathcal{O}(\sqrt{\Delta/\varepsilon})$ is sufficient to guarantee that $\|[_k^\infty\mathcal{Z}]\|_\infty < \varepsilon$. Finally, we bound $\|\mathcal{F}\phi'(s)\|_\infty$. Recall that,

$$\mathcal{F} = \sum_{r=0}^\infty \gamma^r \mathcal{T}^r(\mathcal{T} - \mathcal{Z})$$

Then,

$$\|\mathcal{F}\phi'(s)\|_\infty = \left\| \left[ \sum_{r=0}^\infty \gamma^r \mathcal{T}^r (\mathcal{T} - \mathcal{Z}) \right] \phi'(s) \right\|_\infty$$

$$= \left\| \sum_{r=0}^\infty \gamma^r \mathcal{T}^r (\mathcal{T} - \mathcal{Z}) \phi'(s) \right\|_\infty$$

$$\leq \sum_{r=0}^\infty \gamma^r \| \mathcal{T}^r (\mathcal{T} - \mathcal{Z}) \phi'(s) \|_\infty$$

$$\leq \sum_{r=0}^\infty \gamma^r \| \mathcal{T}^r \|_\infty \| (\mathcal{T} - \mathcal{Z}) \phi'(s) \|_\infty$$

$$\leq \| (\mathcal{T} - \mathcal{Z}) \phi'(s) \|_\infty \sum_{r=0}^\infty \gamma^r \| \mathcal{T}^r \|_\infty$$

We bound $\|(\mathcal{T} - \mathcal{Z})\phi'(s)\|_\infty$ and $\sum_{r=0}^\infty \gamma^r \|\mathcal{T}^r\|_\infty$ separately. By the conditions of the lemma, $\|\phi'_n(s)\| \leq Cn$. Then using the fact that $\|\mathcal{Z}\|_\infty, \|\mathcal{T}\|_\infty < \Delta$.

$$\|(\mathcal{T} - \mathcal{Z})\phi'(s)\|_\infty = \max_{i \in \mathbb{N}} \left| \sum_{j=1}^\infty [\mathcal{T} - \mathcal{Z}]_{ij} \phi'_j(s) \right|$$

$$< \max_{i \in \mathbb{N}} \left| \sum_{j=1}^\infty \frac{2\Delta}{\zeta(3) i^3 j^3} C j \right|$$

$$= \frac{2C\Delta}{\zeta(3)} \left| \sum_{j=1}^\infty \frac{1}{j^2} \right|$$

$$= \frac{2C\Delta\zeta(2)}{\zeta(3)}$$

Then by the conditions of the lemma, $\|\mathcal{T}\|_\infty < \Delta < \frac{1}{2\gamma}$, which implies,

$$\sum_{r=0}^\infty \gamma^r \|\mathcal{T}^r\|_\infty \leq \sum_{r=0}^\infty \gamma^r \|\mathcal{T}\|_\infty^r$$

$$< \sum_{r=0}^\infty \gamma^r \frac{1}{(2\gamma)^r}$$

$$= \frac{1}{1 - \frac{1}{2}} = 2$$

Combining these previous results gives,

$$\|\mathcal{F}\phi'(s)\|_\infty \leq \|(\mathcal{T} - \mathcal{Z})\phi'(s)\|_\infty \sum_{r=0}^\infty \gamma^r \|\mathcal{T}^r\|_\infty < \frac{4C\Delta\zeta(2)}{\gamma\zeta(3)}$$

∎

The next lemma states conditions on the partial derivatives of the transition functions so that the infinite-matrix representations over a trigonometric basis fulfill the conditions of Lemma 10.

**Lemma 11** *If $P(\tilde{s}|s) = \sum_{i,j=1}^\infty \phi_i(\tilde{s}) \mathcal{Z}_{ij} \phi_j(s)$ where $\{\phi_n(s)\}$ are the trigonometric basis functions, and*

$$\left| \frac{\partial^6}{\partial \tilde{s}^3 \partial s^3} P(\tilde{s}|s) \right| < \frac{\pi^6 \Delta}{\zeta(3)},$$

*then,*

$$|\mathcal{Z}_{ij}| < \frac{\Delta}{\zeta(3)i^3 j^3}.$$

## Proof

We defined the basis of trigonometric functions $\{\phi_n(s)\}$ as

$$\phi_n(s) = \cos\left(\lfloor n/2 \rfloor \pi s\right) \quad \text{for all odd } n,$$
$$\phi_n(s) = \sin\left((n/2)\pi s\right) \quad \text{for all even } n,$$

where $\lfloor . \rfloor$ is the floor function.

First we directly represent the partial derivative in terms of the trigonometric basis. We define $\mathcal{Z}'_{ij}$ as

$$\mathcal{Z}'_{ij} \equiv \int_{s_1, s_0 \in S} \frac{\partial^6}{\partial s_1^3 \partial s_0^3} P(s_1|s_0) \phi_i(s_1) \phi_j(s_0) ds_1 ds_0$$

$$\Rightarrow \frac{\partial^6}{\partial s_1^3 \partial s_0^3} P(s_1|s_0) = \sum_{i,j=1}^{\infty} \phi_i(s_1) \mathcal{Z}'_{ij} \phi_j(s_0).$$

We use the bound on the sixth order partial derivative to bound the values of $\mathcal{Z}'$ as follows.

$$|\mathcal{Z}'_{ij}| = \left| \int_{s_1, s_0 \in S} \frac{\partial^6}{\partial s_1^3 \partial s_0^3} P(s_1|s_0) \phi_i(s_1) \phi_j(s_0) ds_1 ds_0 \right|$$

$$< \frac{\pi^6 \Delta}{\zeta(3)} \left| \int_{s_1, s_0 \in S} \phi_i(s_1) \phi_j(s_0) ds_1 ds_0 \right|$$

$$\leq \frac{\pi^6 \Delta}{\zeta(3)}.$$

Next we represent the sixth order partial derivative using the entries of $\mathcal{Z}$ by differentiating the series representation of $P(s'|s)$.

$$P(s_1|s_0) = \sum_{i,j=1}^{\infty} \phi_i(s_1) \mathcal{Z}_{ij} \phi_j(s_0) \Rightarrow$$

$$\frac{\partial^6}{\partial s_1^3 \partial s_0^3} P(s_1|s_0) = \sum_{i,j=1}^{\infty} \frac{\partial^3}{\partial s_1^3} \phi_i(s_1) \mathcal{Z}_{ij} \frac{\partial^3}{\partial s_0^3} \phi_j(s_0).$$

If $n$ is odd then,

$$\frac{\partial^3}{\partial s^3} \phi_n(s) = \frac{\partial^3}{\partial s^3} \cos(\lfloor n/2 \rfloor \pi s) = (\pi n)^3 \sin(\lfloor n/2 \rfloor \pi s) = (\pi n)^3 \phi_{n-1}(s).$$

If $n$ is even then,

$$\frac{\partial^3}{\partial s^3} \phi_n(s) = \frac{\partial^3}{\partial s^3} \sin((n/2)\pi s) = -(\pi n)^3 \cos((n/2)\pi s) = -(\pi n)^3 \phi_{n+1}(s).$$

Therefore we can map the entries of $\mathcal{Z}$ to $\mathcal{Z}'$. The exact mapping is not important since all entries of $\mathcal{Z}'$ are bounded by $\frac{\pi^6 \Delta}{\zeta(3)}$. Finally, we have,

$$\pi^6 i^3 j^3 |\mathcal{Z}_{ij}| = |\mathcal{Z}'_{i\pm 1, j\pm 1}| < \frac{\pi^6 \Delta}{\zeta(3)}$$

$$\Rightarrow |\mathcal{Z}_{ij}| < \frac{\Delta}{\zeta(3)i^3 j^3}.$$

∎

Combining the statements of Lemma 10 and Lemma 11 along with the fact that $|\frac{d}{ds}\cos(\lfloor n/2\rfloor \pi s)|, |\frac{d}{ds}\sin((n/2)\pi s)| \le \pi n$ constitutes the proof of Lemma 7.

We can strengthen the assumption on $\Delta$ in Lemma 7 from $\Delta < \frac{1}{2\gamma}$ to $\Delta < \frac{1}{4\gamma}$ to achieve a simple sample complexity for Algorithm 2 which only depends on the probability of failure $\delta$ and the separability measure $\beta$.

**Corollary 12** *If the IRL problem is $\beta$-separable and*

$$\left| \frac{\partial^6}{\partial \tilde{s}^3 \partial s^3} P_a(\tilde{s}|s) \right| < \frac{\pi^6}{4\gamma\zeta(3)}, \ \forall a \in A,$$

*then with $\mathcal{O}\left(\frac{1}{\beta^3}\log\frac{1}{\beta\delta}\right)$ samples, Algorithm 2 with a trigonometric basis outputs a correct reward function $R$ with probability at least $1 - \delta$.*

**Proof**

By Lemma 7, we have the following bounds when substituting in $\Delta < \frac{1}{4\gamma}$.

$$(i)\ \|\mathcal{F}\phi'(s)\|_\infty < \frac{\pi\zeta(2)}{\gamma\zeta(3)}, \quad (ii)\ \|\mathcal{Z}\|_\infty < \frac{1}{4\gamma}, \quad (iii)\ \|[{}_k^\infty \mathcal{Z}]\|_\infty < \varepsilon, \ \text{with } k \in \mathcal{O}\left(\sqrt{\frac{1}{\varepsilon}}\right)$$

Theorem 6 then guarantees that Algorithm 2 will return a correct reward function with probability at least $1-\delta$ for some $\gamma \in (0,1)$, $c > 0$, $k \in \mathcal{O}\left(\sqrt{\frac{1}{\varepsilon}}\right)$, and $n \in \mathcal{O}\left(\frac{1}{(\beta - c\rho)^2}k^2 \log\frac{k}{\delta}\right)$.

To see this, let

$$\varepsilon = \frac{\beta - c\rho}{8}\left(\frac{1}{2} - \gamma\Delta\right)^2 = \frac{\beta - c\rho}{128}, \quad \text{and} \quad \rho = \frac{\pi\zeta(2)}{\gamma\zeta(3)}.$$

We can simplify the sample complexity result by setting $c$ such that $c\rho = \frac{\beta}{2}$ and using $k \in \mathcal{O}\left(\sqrt{\frac{1}{\varepsilon}}\right)$ to obtain,

$$\mathcal{O}\left(\frac{1}{(\beta - c\rho)^2}k^2 \log\frac{k}{\delta}\right) = \mathcal{O}\left(\frac{1}{(\beta - c\rho)^2}\frac{1}{\beta - c\rho}\log\frac{1}{(\beta - c\rho)\delta}\right)$$

$$= \mathcal{O}\left(\frac{1}{(\beta)^2}\frac{1}{\beta}\log\frac{1}{\beta\delta}\right)$$

$$= \mathcal{O}\left(\frac{1}{\beta^3}\log\frac{1}{\beta\delta}\right).$$

which concludes the proof. ∎

# C  Experiments

In order to verify our theoretical results, we test our algorithm on simple randomly generated IRL problems. To accomplish this, we randomly generate IRL problems with polynomial transition functions. Using polynomial transition functions has two main advantages. First, we can obtain simple closed form solutions when generating the coefficient matrix $\mathcal{Z}^{(a)}$. Second, the transition function is infinitely differentiable, meaning the truncation error rapidly decreases with $k$. We conducted our experiments on a 64-core AMD Epyc 7662 'Rome' processor with 256 GB memory and coded our experiments using Python 3.

## C.1  Generating transition functions

First, we describe a way to generate a random polynomial which is a valid probability distribution function over $S = [-1, 1]$. Let $\mathcal{P}_r = a(x - b)^{2r}$ where $a, b \sim$Uniform$(0, 1)$ denote a polynomial with variable $x$. Notice that for all $r \in \mathbb{N}$, $\mathcal{P}_r$ is non-negative over $S$. Therefore, we can construct a

non-negative (even degree) polynomial $\mathcal{P} = \sum_{r=1}^{d/2} \mathcal{P}_r$. Re-normalizing $\mathcal{P}$ so that it integrates to $1$ over $S$ then makes $\mathcal{P}$ a valid probability density function since it integrates to one and is non-negative.

To generate a transition function, $P(s'|s)$, create two random polynomial distributions $\mathcal{P}_a$ and $\mathcal{P}_b$ as described above. Then let $P(s'|s) = (1 - s^2)\mathcal{P}_a(s') + s^2\mathcal{P}_b(s')$. For each fixed $s$, $P(s'|s)$ is a weighted average of two probability density functions, and is therefore a probability density function.

## C.2 Sampling the coefficient matrix (Algorithm 1) (Figure 2)

In order to test Algorithm 1 and the guarantees of Theorem 3, we must compute $\mathcal{Z}$ and $\widehat{\mathcal{Z}}$. Specifically, to create Figure 2, we first generate a fixed random transition function. Next, we implement Algorithm 1 to compute $\widehat{\mathcal{Z}}$ for a specified truncation size $k$, where we compute the one-step transition $s'_r \sim P_a(\cdot|\bar{s}_r)$ using inverse transform sampling with eight bits of precision. We cannot compute the infinite matrix $\mathcal{Z}^{(a)}$ completely, so instead we restrict ourselves to measuring the error $\|[^k\mathcal{Z}] - \widehat{\mathcal{Z}}\|_\infty$. The $i,j$-th entry of $\mathcal{Z}^{(a)}$ is given by the following formula.

$$
\begin{aligned}
\mathcal{Z}_{ij} &= \int_{-1}^{1} \int_{-1}^{1} ((1 - s^2)\mathcal{P}_a(s') + s^2\mathcal{P}_b(s'))\phi_i(s')\phi_j(s)ds'ds \\
&= \int_{-1}^{1} \left((1 - s^2)\int_{-1}^{1} \mathcal{P}_a(s')\phi_i(s')ds'\right)\phi_j(s)ds + \int_{-1}^{1}\left(s^2\int_{-1}^{1}\mathcal{P}_b(s')ds'\phi_i(s')\right)\phi_j(s)ds.
\end{aligned}
$$

We then use the fact that a polynomial is a sum of monomials. This allows us to give a simple recursive specification of the integral of a polynomial with each trigonometic basis function using integration by parts.

## C.3 Solving the IRL problem (Algorithm 2) (Figure 3)

Next, we conduct experiments testing our ability to solve a randomly generated IRL problem. In these experiments, we use $\gamma = 0.7$ and $|A| = 3$, where each transition function is as described above. We compute each $\widehat{\mathcal{Z}}^{(a)}$ using Algorithm 1 and the compute the reward vector $\widehat{\alpha}$ using Algorithm 2. We classify the returned $\widehat{\alpha}$ as "correct" or "incorrect" by checking if $\widehat{\alpha}^T \widehat{\mathcal{F}}^{(a)}\phi(\bar{s}) > 0$ for all $\bar{s} \in \bar{S}$ where $\bar{S}$ is a covering set over $[-1, 1]$ of size 100. If this inequality does not hold for any $\bar{s} \in \bar{S}$, then the reward vector is classified as "incorrect". To generate Figure 3, we repeat the above process 320 times for each value of $n$ and each value of $k$.

Additionally, we verified that we implement Algorithm 2 correctly by checking the empirical expected reward on a set of 100 points in $[-1, 1]$ for several IRL problems. To compute the empirical expected reward at state $s_0$, we sample a sequence of point $s_1...s_6$ such that $s_r \sim P_a(\cdot|s_{r-1})$ using inverse transform sampling, as described above. We average the discounted reward of these 6 points over 6000 samples and find that we succeed in generating a reward vector $\widehat{\alpha}$ such that the expected reward of the first action is higher than the expected reward of any other action at each starting point $s_0$.

## C.4 Effect of $\beta$-separability

As expected from the sample complexity result of Theorem 6, we found the $\beta$-separability had a significant impact on the sample complexity of Algorithm 2 (see Figure 4 below).

We find the separability measure $\beta$ of a randomly generated IRL problem by running Algorithm 2 using the exact $[^k\mathcal{Z}]$ matrices with $k = 11$. Using the exact matrices removes any error introduced by sampling, and since the transition functions are infinitely differentiable, the truncation error is minimal. Since $\widehat{\alpha}\mathcal{F}^{(a)}\phi(s) \geq 1$ holds approximately, we can approximate $\beta$ as $\beta = \frac{1}{\|\widehat{\alpha}\|_\infty}$.

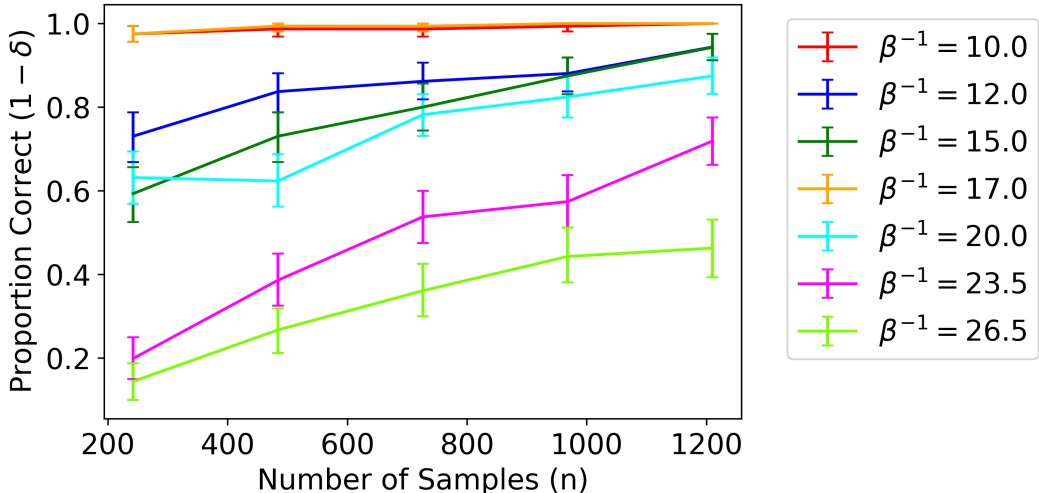

Figure 4: This graph shows the proportion of out of of 160 trials at each value of $n$ where the returned reward vector is approximately Bellman optimal versus the number of samples. We plot the results from multiple generated IRL problems across a range of values for $\beta^{-1}$. We observe that $\beta$ has a significant impact on the samples needed in Algorithm 2. Error bars represent 95% confidence and are computed by bootstrapping out of 120 trials.