# OpenReview forum: "Inverse Reinforcement Learning in a Continuous State Space with Formal Guarantees"
_NeurIPS.cc/2021/Conference — NeurIPS 2021 Poster_

### Official Review · Reviewer_aTgp · 2021-07-11

**Rating:** 6
**Confidence:** 3

**Summary:**

The authors present the first algorithm for IRL in continuous setting with formal guarantees including sample and computational complexity bounds. The theoretical analysis is based on the theory of infinite matrices. The theoretical bounds are corroborated by synthetic experiments.


**Limitations And Societal Impact:**

yes

**Main Review:**

The ideas of the proofs and setups are extended from [Ng and Russell, 2000] and [Komanduru and Honorio, 2019]. Although the technique used in the proofs -- theory of infinite matrices --  is not new, (the book referred to in the text dated back to 1950), applying it in the IRL context seems novel and inspiring. Furthermore, the algorithm proposed seems to be the first IRL algorithm (i.e. learning reward function given optimal policy and query access to transition function) in continuous setting that has formal guarantees.

The impact the work can be boarder if the authors discuss a connection to state-of-the-art IRL and model-based RL algorithms and lift the assumptions that not unusually satisfied in practice, such as
- the demonstration policy is known and is optimal and deterministic
- the optimal reward is unique and \beta separable, which can be violated sometimes (see upper left of Page 4 [3])
- the MDP can be reset to any state

The paper is very well-written and easy to follow. However, I think there’s still some room for improvement. For example, Eq. (6) is very important, but it lacks explanation and intuition. It would help readers by providing some context beyond mentioning that it’s an analogue to the equations of Ng and Russel.


[1] Sun, Wen, et al. "Provably efficient imitation learning from observation alone." International conference on machine learning. PMLR, 2019.

[2] Abbeel, Pieter, and Andrew Y. Ng. "Exploration and apprenticeship learning in reinforcement learning." Proceedings of the 22nd international conference on Machine learning. 2005.

[3] Ng, Andrew Y., and Stuart J. Russell. "Algorithms for inverse reinforcement learning." ICML. Vol. 1. 2000.

Other comments:

I wonder whether the theoretical results in [Komanduru and Honorio, 2019] can be applied to continuous settings simply by discretizing the continuous state space. For example, this is done in Section 5.1 Lipschitz Continuous MDP case study in [1].

I think this paper is related to a line of research on imitation learning that learns the transition dynamics from demonstration, e.g. [2].

This work extends from discrete setting to continuous state space. Is the same technique able to further extend to continuous action space?

Other minor comment:

In Fig. 1., it seems that the blue curve is a shifted-down version of the red curve. Is this always the case?

In the caption of Fig. 3 \beta = 26.5. And in the legend of Fig. 4 in the Appendix C.4., \beta^{-1} = 26.5. Is there a typo?

The number of trials in the experiments is missing.


**Time Spent Reviewing:**

4 hours

---

> ### Author Response · Authors · 2021-08-11
> **Reply to initial review by Reviewer aTgp**
>
> Thank you for your detailed review.  We appreciate your acknowledgment of the novelty of our approach, the primality of our theoretical results, and the clarity of our writing.  We would like to clarify that, while the formalism of infinite-matrices is not new, our results leverage exciting new results in theoretic understanding of IRL. We would like to clarify a few points to address your concerns.
>
> >"The impact the work can be boarder if the authors discuss a connection to state-of-the-art IRL and model-based RL algorithms and lift the assumptions that not unusually satisfied in practice, such as: 1) the demonstration policy is known and is optimal and deterministic. 2) the optimal reward is unique and \beta separable , which can be violated sometimes (see upper left of Page 4 [3]). 3) the MDP can be reset to any state." (Our numbering and formatting)
>
> We argue that both assumptions (1) and (3) that you mention are standard in theoretical analysis of IRL, and they allow us to focus on our core contributions. In fact, we are unaware of any theoretically guaranteed method for IRL with unknown non-parametric transition dynamics even in the discrete case which does not use assumption (3).  Similarly, we are unaware of any work even in the discrete case which does not de facto use assumption (3). For a more detailed comparison to related work with these assumptions, please refer to our response to Reviewer pRjN starting with the phrase: "The assumption that we can sample."
>
> Regarding the second concern, **we do not assume that the optimal reward is unique**.  Rather, we resolve the ambiguity of choosing the optimal reward function by applying L1-regularization to the reward series representation.  This approach is particularly appealing since this provides a sparse representation of the optimal reward.  The paper you cite by Ng & Russell handles the ambiguity in a similar way, but their method was proven to often not return a correct reward function by [Komanduru and Honorio, 2019].  Furthermore, the assumption of $\beta$-separability is separate from the question of a unique reward function.  Strict separability assumptions are common in statistical learning theory, and, as argued in [Komanduru and Honorio, 2019], the IRL problem is not well-posed without it.  We also note that our experiments (see Appendix C.4) support the need to consider $\beta$-separability as it has an important impact on observed sample complexity.
>
> >"The paper is very well-written and easy to follow. However, I think there's still some room for improvement. For example, Eq. (6)  is very important, but it lacks explanation and intuition."
>
> We agree with this observation.  We will add more of the intuition for Eq. (6) from the proof in the appendix to the main text on how Eq. (6) follows by representing expected reward through matrix multiplication in the Q-function and we will preface its use in Lemma 2 as a sufficient condition for optimality.
>
> >"I wonder whether the theoretical results in [Komanduru and Honorio, 2019] can be applied to continuous settings simply by discretizing  the continuous state space. For example, this is done in Section 5.1 Lipschitz Continuous MDP case study in [1]."
>
> While we believe it would be possible to apply the results of Komanduru and Honorio using a discretization approach, we do not believe such an analysis would be straightforward or preferable to ours.  In fact, we believe that our approach **using a series representation over an orthonormal function basis leads to a more conceptually intuitive formalization, which in turn, allows simpler proofs and  stronger results**.  We illustrate these claims below.
>
> One example which highlights that our approach is more conceptually intuitive is our definition of $\beta$-separability (Definition 2). An MDP is $\beta$-separable for some $\beta$ if there exists a reward function such that the Q-function of the optimal action is at least $\beta$ greater than the Q-function of a non-optimal action for all states in the state space.  We also know that we can approximate such a reward function to arbitrary accuracy using a truncated representation of the function series, which guarantees the statistical separability of the optimal and non-optimal policies in our algorithm at a finite truncation size $k$. **Trying to apply the definition of $\beta$-separability in the Komanduru paper (Definition 4.1) using discretization leads to a much less appealing formalism**.  Notice that for any transition function that is bounded and continuous, the probability of transitioning from a point in bin $B_i$ to bin $B_j$ is $O(1/k)$, where $k$ is the number of bins in the discretization.  Using the triangle inequality and Holder's inequality, we can conclude that $\beta$-separability of a discretized problem decreases with the number of bins $k$ at a rate of $O(1/k)$.  For one, this is unappealing because the $\beta$-separability in the limit of the discretization is zero for all bounded MDPs, which diminishes the value of $\beta$ in describing the problem.  Secondly, this means that by Theorem 5.2, **the Algorithm of Komanduru will require $O(k^4)$ samples where $k$ is the representation size of the discretized problem. Meanwhile, our sample complexity scales with $O(k^2)$, where $k$ is the length of the truncation**. One may be tempted to "normalize" the definition of $\beta$-separability with $k$ bins by allowing the L1-norm of the reward vector to grow at a rate of $O(k)$, but we remind that this would then invalidate Komanduru's concentration analyses of the estimated matrices.  Furthermore, it is not even conceptually obvious that the $O(k^2)$ gap in sample efficiency between our method and the discretized approach could be removed.  Notice that **the work of Komanduru gives no way to encode distance between states, meaning transition probabilities between discrete states are arbitrary.  However, our truncation method automatically leverages the fact that the transition function is smooth**, since lower term basis functions generally vary slowly.  As an aside, this type of behavior motivates the field of "functional data analysis", which is based on the idea that treating continuous data as discrete loses information and produces worse results.
>
> We've given the example of $\beta$-separability to show the technical complications and conceptual challenges in handling the continuous state space setting through discretization.  Although, we would like to emphasize that such problems would show up throughout the analysis, such as when accounting for the propagation of discretization error when computing the $F$ matrices or setting up the linear program to regularize the returned reward.  However, even beyond these challenges in implementing the discretization approach, our method using orthonormal bases is appealing.
>
> **Orthonormal function bases have been extensively studied, and such work allows us to conclude stronger guarantees easily**.  For example, if the transition function is known to be infinitely continuously differentiable, then it is known that the **truncation error will decrease at the rate of $O(1/\exp(k))$** over a trigonometric basis.  Meanwhile, the discretization you provide in citation [1] only guarantees that **discretization error will decrease at a rate of $O(1/k)$**.  Another benefit is that for transition functions which are smooth and continuous, it is more believable that it would have a sparse or near-sparse function series representation than a sparse discretization.  Such results appear in other fields of computer science, such as the use of a wavelet basis for sparse image representation.  This in turn makes our use of L1-regularization to return **a reward function with a sparse series representation more sensible than trying to return a sparse piecewise reward function**.  We must omit several more advantages of our approach over discretization due to space limitations, but we are happy to elaborate if desired.
>
> >"This work extends from discrete setting to continuous state space. Is the same technique able to further extend to continuous action space?"
>
> We believe that our work provides a useful stepping stone toward achieving guaranteed algorithms for the continuous state-action setting.  One reason for this is that our estimator of the transition functions can be easily extended to this setting under smoothness assumptions.  Additionally, effective reinforcement learning algorithms in the continuous state-action setting were developed from theoretic work on RL in the continuous state and discrete action setting.  Due to space limitations, we refer to our other responses for more details.
>
> For more detail on the described potential extension of our method to the continuous state-action setting, please see the response to Reviewer MeZM starting with the phrase: "We believe that we could potentially extend our approach."
>
> For more detail on the line of research which built off of Probably Approximately Correct (PAC) learning in RL over a continuous state discrete action setting, please see the response to Reviewer pRjN starting with the phrase: "Because of the lack of prior work."
>
> << Minor Comments >>.  Regarding your minor comments, we thank you for pointing out the typos.  Regarding Fig. 1, the red and blue lines do not necessarily have the same shape.

---

> > ### Comment · Reviewer_aTgp · 2021-08-15
> > **Response to author's response**
> >
> > Thank you for your detailed response and the clarification on why the proposed method is preferred over discretizing the continuous state space. I have the following two questions and would appreciate your thoughts.
> >
> > (1) In your response to Reviewer MtyF and Review pRjN, you mentioned that in d-dimension setting, your method does not require any additional assumptions. However, the theorem in Section 7 (Theorem 8) assumes the decomposition of probability transition functions. Would you please clarify that?
> >
> > (2) "Regarding the second concern, we do not assume that the optimal reward is unique. Rather, we resolve the ambiguity of choosing the optimal reward function by applying L1-regularization to the reward series representation." If we consider the reward vectors with L1-norm being 1, does Def. 2 assume uniqueness of the best reward vector and the second best reward vector is at least \beta worse than the best one? Would you please point me to the literature on statistical learning theory where strict separability assumptions are necessary? I thought those assumptions can improve the learning rate but are not necessary. Thank you.

---

> > > ### Author Response · Authors · 2021-08-19
> > > **Response to Reviewer aTgp**
> > >
> > > Thank you for your comments.
> > >
> > > (1)
> > >
> > > We clarify that when we say our method can be generalized to the $d$-dimensional setting without additional assumptions, we are referring to our discussion on Lines 257-264.  You are correct that Theorem 8 assumes the transition function can be decomposed.  Our goal in Section 7 was to briefly demonstrate the flexibility of our infinite matrix formulation.  However, given that several authors have raised questions about this section, we would like to propose a slightly more sophisticated extension to the $d$-dimensional setting.
> > >
> > > Our new extension would guarantee the correctness of our method in the case where the $d$-dimensional transition functions of the given MDP can be decomposed into the product of $T$ factor transition functions where the $i$-th factor operates on a subspace of the total state space, $S^{(i)} \subset S$, such that $\dim(S^{(i)}) \leq q$ and $S = S^{(1)} \times S^{(2)} \times ... \times S^{(T)}$. In other terms, $P(s_1 | s_0) = \prod_{i=1}^T P_i(s_1^{(i)} | s_0^{(i)})$ and each component transition function in the product is over a state space of at most dimension $q$. Using our current results with a few slight modifications, we can guarantee a solution to this problem with $O(T \exp(q))$ scaling to our current complexity results.
> > >
> > > This formulation encompasses both approaches we currently discuss in Section 7.  Notice that if we use one factor of $d$-dimensions ($T = 1$ and $q = d$), then we get the $\exp(d)$ scaling method that makes no additional assumptions on the transition functions, i.e., the original $d$-dimensional problem.  If we decompose the transition function as a product of $d$ one-dimensional transition functions ($T = d$ and $q = 1$), then we recover the same conclusion as Theorem 8, i.e., we can solve the total IRL problem by solving $d$ one-dimensional IRL problems.  We briefly describe what we would change for this conclusion.
> > >
> > > We first expand on Lines 257-264 of our paper (the T=1, q=d case) and explain in more detail how one can solve a $d$-dimensional IRL problem without any additional assumptions. To start with, we would change our orthonormal basis functions to be over a $d$-dimensional space such that $\phi_i(s):[-1, 1]^d \rightarrow [-1, 1]$.  This can be done by letting each $\phi_i$ to be the product of $d$ one-dimensional basis functions.  The key insight is that our proofs rely on representing $\int_{s_1\in S} P(s_2 |s_1) P(s_1|s_0) ds_1$ as a product of infinite matrices and the expectation $\int_{s_1 \in S} R(s_1) P(s_1|s_0) ds_1$ as an infinite matrix-vector product.  Since this holds equally well when $S$ is $d$-dimensional, we don't need to change any of our results prior to Lemma 5.  In Lemma 5, we use a $c$-cover, $\bar{S}$, of $[-1, 1]$ and $\rho$-Lipschitz continuity of $\widehat{\alpha} \widehat{\mathcal{F}} \phi(s)$ to lower bound the Bellman optimality of our returned reward function.  This same argument holds by using a $c$-cover of the $d$-dimensional state space and $\rho$-Lipschitz continuity of $\widehat{\alpha}^\top \widehat{\mathcal{F}} \phi(s)$ with respect to the $\ell_2$-norm. Guaranteeing that $\alpha^\top \mathcal{F} \phi(s)$ is Lipschitz in the $d$-dimensional setting is no more difficult than guaranteeing it in the one-dimensional setting, since guaranteeing that it is $\frac{\rho}{\sqrt{d}}$-Lipschitz in each dimension guarantees it is $\rho$-Lipschitz with respect to the $\ell_2$-norm.  This part on creating a $c$-covering for the $d$-dimensional state space is where the $\exp(d)$ computational scaling will occur, since the covering number of a $d$-dimensional hypercube is $\exp(d)$, we will need $\exp(d)$ points in our cover and hence $\exp(d)$ constraints in the linear program we solve. However, the proof remains the same.
> > >
> > > The above modifications shows that our method can be used to solve a $d$-dimensional IRL problem in $\exp(d)$ time without additional assumptions.  To extend our method to general values of $T$ and $q$, notice that our proof of Theorem 8 only assumes that the transition functions of an MDP can be decomposed, not that component transition functions are one-dimensional.  Therefore, we can immediately conclude by the proof of Theorem 8 that we can solve a $d$-dimensional IRL problem by solving each of the $T$ IRL problems in the decomposition in $\exp(q)$ time.
> > >
> > >
> > > (2)
> > >
> > > The definition of $\beta$-separability does not assume that the reward function is unique, it does not assume that the reward function considered in the definition is the “best” in any sense, and it only considers one reward function, so there is no “second best”.  We would like to carefully explain the intuition of the $\beta$-separability assumption and why it is necessary to make these points clear.
> > >
> > > We refer to Section 3 of [Komanduru and Honorio, 2019] to explain why a strict separability assumption is needed.  Through a geometric perspective, the authors show that every discrete IRL problem can be classified into three regimes, Regime 1 where there is no non-zero reward to make the given policy optimal, Regime 2, where the given policy can be made optimal but not strictly optimal, and Regime 3, where the given policy can be made strictly optimal. In Figure 2, they show that problems in Regime 2 will fall randomly into Regime 1 or Regime 3 due to error in estimating the transition dynamics.
> > >
> > > Here we provide an even simpler example on why strict separability is needed.  Consider a two-state MDP such that the probability of transitioning to state 1 after taking action 1 is $1/2 + \epsilon$ in both states.  Similarly, let the probability of transitioning to state 2 after taking action 2 equal $1/2 + \epsilon$ in both states.  In formulas: $P_{a_1}(s_1 = s^{(1)} | s_0 = s) = 1 + \epsilon$ and $P_{a_2}(s_1 = s^{(2)} | s_0 = s) = 1 + \epsilon$.
> > >
> > > Clearly, we can make $\pi \equiv a_1$ optimal by setting the reward of state 1 to be higher than of state 2.  However, if $\epsilon$ is small, then we could easily estimate that the probability of transitioning to state 1 by action 1 is less than $1/2 -  \epsilon$ and transition to state 2 by action 2 is less than $1/2 - \epsilon$.  This would cause us to return an incorrect reward function by setting reward of state 2 to be higher than the reward of state 1.  This motivates our assumption, because if allow $\epsilon$ to be arbitrarily small, then we cannot guarantee we estimate the transition dynamics to sufficient accuracy with high probability for any finite number of samples.
> > >
> > > Given this motivation for a strict separability assumption, we would like to give some intuition on our definition of $\beta$-separability.  We call a problem $\beta$-separable if there exists a reward function with infinite-vector representation $\alpha$ such that $\alpha^\top \mathcal{F}^{(a)} \phi(s) \geq \beta > 0$ and $\|\alpha\|_1 = 1$ for all $a \in A$ and $s \in S$.  We break this statement down.  First, recall that $\alpha^\top \mathcal{F}^{(a)} \phi(s) \geq \beta$ is equivalent to saying that $Q(s, a_1) - Q(s, a) \geq \beta$ for the reward function represented by $\alpha$, i.e., the reward function is $\beta$-strictly Bellman optimal.  As we argued before, if the Bellman criteria is not fullfilled strictly, then we cannot guarantee correctness with high probability.  Second, we impose that $\|\alpha\|_1 = 1$ because we can arbitrarily scale the difference of the Q-functions by scaling the reward functions, so we must control this by fixing the scale.  Third, notice that we do not mention an "optimal" reward function or maximize any value in the definition.  We simply say that a reward function fulfilling this criteria exists.  A given IRL problem can be $\beta$-separable for infinitely many values of $\beta$.  One attains the best complexity results by maximizing $\beta$, but it is not necessary to do so.
> > >
> > > To recap your questions, we do not assume that there exists a reward function that is unique or optimal in any way.  We merely assume that we know there exists *some* reward function so that there is *at least some* strict separation $\beta$ between the Q-functions of the optimal and non-optimal policies under such a reward, i.e., there exists a $\beta$-strictly Bellman optimal reward function.  As we showed above, if we do not assume a lower bound on the separation, then we can never guarantee we have estimated the transition dynamics accurately enough to solve the IRL problem with high probability. Please let us know if any of the above statements are unclear to you.  Thank you.

---

> > > > ### Comment · Reviewer_aTgp · 2021-08-21
> > > > **Thank the authors for their reply**
> > > >
> > > > I would like to think the authors for their very helpful reply.

---

### Official Review · Reviewer_MeZM · 2021-07-12

**Rating:** 6
**Confidence:** 3

**Summary:**

The paper studies the IRL problem in the presence of a compact state space and finite actions. Under the assumption that the transition model of the MDP can be represented by means of an (infinite) set of orthogonal basis functions, the authors prove that there always exists a reward function explaining the expert's behavior that can be constructed using the same basis functions. Then, the paper proposes an approach for estimating a reward function endowed with sample complexity guarantees. Synthetic experiments are then provided and the paper concludes with a discussion for the extension to multiple dimensions.

**Limitations And Societal Impact:**

The paper is mainly theoretical, thus, I do not foresee any societal impact.

**Main Review:**

***Major***
1. (Related Works) I think that the paper lacks some important related works. First, in line 26, the authors claim that "Existing algorithms for IRL in the continuous setting generally do 27 not directly solve the IRL problem, but instead take a policy matching approach". However, there exists a class of IRL methods, called "Truly-batch model-free" that actually address the IRL problem with continuous states (and also continuous actions) producing as output a reward function [1, 2]. Second, the authors claim tha  "there is a lack of formal guarantees for IRL methods". Very recently, an IRL algorithm with sample complexity analysis has been presented [3]. Third, the idea of using an orthogonal basis in IRL was first introduced by [4].

2. The authors consider in Definition 1 a state-only reward function. Is the presented approach bounded to that assumption, or can it be extended to reward functions depending on state-action pairs?

3. (Significance of the Theoretical Results) I have some concerns about the significance of the theoretical results about sample complexity presented on the paper, especially about how much the assumptiona are realistic. I will be more precise.
- Theorem 3. In order to get the concentration result, it is assumed that the complement of the k-truncation of the infinite matrix Z has a norm bounded by \epsilon. Is this assumption realistic? Isn't Z estimated from samples, so how can it be enforced? It seems to me that this represents just a way of computing a union bound over a finite (k^2) number of entries. Can the authors clarify this point?
- Lemma 4. This lemma holds under the assumption that the \infty norm of the infinite matrix \hat{Z} is bounded by 1/\gamma? How is this assumption guaranteed in practice, given that \hat{Z} is estimated? More in general, which is the intuition behind the truncation of the infinite matrix? How is k selected?
- Lemma 5. Also here, the bounds on the \infty-norm of matrix \mathcal{F} as well as the \infty-norm of the difference between \mathcal{F}  and \hat{\mathcal{F}}. How are they guaranteed to hold?

In general, I found difficulties in getting convinced on the assumptions used in the presented results and quite abstract. I would greatly appreciate that the authors motivate/justify all the assumptions and show how/when they are satisfied in practice, based on the properties of the problems at hand. The same holds for Section 5. How are the conditions of Lemma 7 guaranteed? Is there a particular class of MDPs in which it is satisfied?

- Theorem 8. The authors show that when considering multiple dimensions, the IRL problem, as presented, becomes tractable when the reward function decomposes linearly over the dimensions. I think this assumption is quite restrictive. Think, for instance, to a two-dimensional gridworld where the goal is to reach a certain goal cell. The sparse reward that assigns +M to the goal cell and 0 elsewhere cannot be decomposed as in Theorem 8. Can the authors elaborate on this point?


***Minor***
- Footnotes should appear after the punctuation sign.
- Equations (1) and (2). The two expectations are taken wrt different probability distributions. This should be clarified, maybe, introducing proper symbols.
- Algorithms and plots should appear at the top of the page.
- Figure 2: number of trials and meaning of the confidence bars missing.
- Figure 3: the number of runs missing.


***Overall***
I think that the paper provides interesting ideas on a relevant topic (IRL in continuous state spaces). However, I found the results quite abstract and I had trouble understanding how much the assumptions are reasonable and when they are verified in practice. I would greatly appreciate it if the authors reorganize the core of the paper, focusing on clarity. Currently, I opt for a borderline score.


[1] Pirotta, Matteo, and Marcello Restelli. "Inverse reinforcement learning through policy gradient minimization." In Thirtieth AAAI Conference on Artificial Intelligence. 2016.
[2] Ramponi, Giorgia, Amarildo Likmeta, Alberto Maria Metelli, Andrea Tirinzoni, and Marcello Restelli. "Truly batch model-free inverse reinforcement learning about multiple intentions." In International Conference on Artificial Intelligence and Statistics, pp. 2359-2369. PMLR, 2020.
[3] Metelli, Alberto Maria, Giorgia Ramponi, Alessandro Concetti, and Marcello Restelli. "Provably Efficient Learning of Transferable Rewards." In International Conference on Machine Learning, pp. 7665-7676. PMLR, 2021.
[4] Metelli, Alberto, Matteo Pirotta, and Marcello Restelli. "Compatible reward inverse reinforcement learning." In The Thirty-first Annual Conference on Neural Information Processing Systems-NIPS 2017. 2017.

***Post Discussion***
I thank the authors for the detailed feedback. I think that my concerns have been addressed, thus I am raising my score. Regarding the reorganization of the paper, I believe that clarity would benefit from stating all the assumptions of Theorem 6 at the beginning of Section 3 and specify that the ones employed in the intermediate results are a consequence of those.

**Time Spent Reviewing:**

2.5

---

> ### Author Response · Authors · 2021-08-11
> **Reply to initial review by Reviewer MeZM**
>
> Thank you for your detailed review.  We appreciate your acknowledgment of the importance of IRL and the novelty of our ideas.  We would like to clarify a few points to address your concerns.
>
> >"I think that the paper lacks some important related works....."
>
> Regarding the Truly-batch model-free papers [1, 2], we believe these citations could be of interest to a potential reader, so we will mention them in our introduction as examples of methods in the continuous setting which recover an optimal reward, albeit without guarantees.
>
> Regarding the recent IRL algorithm with sample complexity analysis [3], we appreciate you bringing our attention to this paper and will add it to our introduction.  Although, we would like to clarify that it is restricted to a finite state space and was released after the NeurIPS submission deadline, hence contemporaneous to our paper.
>
> Regarding the prior use of orthogonal bases in IRL, we will also mention this paper in our introduction as an example of using orthogonal basis functions in a heuristic-based IRL method.
>
> >"The authors consider in Definition 1 a state-only reward function. Is the presented approach bounded to that assumption, or can it be extended to reward functions depending on state-action pairs?"
>
> We believe that we could potentially extend our approach to a continuous action space with a reward function depending on state-action pairs. We can immediately extend our estimator for the transition function $P_a(s' | s)$ to to an estimator for transition dynamics of the form $P(s' | s, a)$ depending continuously on $a$.  The representation of this continuous transition function over an orthonormal basis would have a representation as an infinite third-order tensor that is analogous to our infinite-matrix representation.  Then, we could represent a reward function depending on the state-action pairs by an infinite-matrix over an orthonormal basis.  We anticipate that we could then represent the expected reward under a policy by contracting the infinite-tensor representing the transition dynamics, the infinite-matrix representing the reward function, and an infinite-vector representing a given policy.  We argue that this will hold since our argument in the infinite-matrix case relies on the fact that the integral of the product of basis functions is one when the functions are the same and zero otherwise.  This still holds in the higher-order case described above.  Our description of such an approach is speculative, since working out the details of the error analysis would constitute an entirely new project.  However, we believe our work can provide guidance for developing methods for the continuous state-action setting where the reward function depends state-action pairs.
>
> >"Theorem 3. In order to get the concentration result, it is assumed that the complement of the k-truncation of the infinite matrix Z has a norm bounded by \epsilon. Is this assumption realistic ? Isn't Z estimated from samples, so how can it be enforced? It seems to me that this represents just a way of computing a union bound over a finite (k^2) number of entries. Can the authors clarify this point?"
>
> The matrix $Z$ is not estimated from samples.  The estimated matrix is represented by $\widehat{Z}$.  The condition that the "complement of the $k$-truncation of the infinite matrix $Z$ has a norm bounded by $\epsilon$ only depends on the true transition functions of the MDP and can be easily verified from the properties of the MDP and basis functions.  For example, enforcing that the series representation of the transition functions (Equation 4) is absolutely convergent is sufficient to guarantee such a value for $k$ exists.  We next discuss the intuition of this point more in response to your following remark.
>
> >"Lemma 4. This lemma holds under the assumption that the \infty norm of the infinite matrix \hat{Z} is bounded by 1/\gamma? How is this assumption guaranteed in practice, given that \hat{Z} is estimated? More in general, which is the intuition behind the truncation of the infinite matrix? How is k selected?"
>
> We show how this assumption is guaranteed in proof of Theorem 6, on line 467 specifically.  We point out that our main theorem, Theorem 6,  does not depend on any estimated values.  It only depends on the properties of the true MDP.
>
> To understand the intuition of our method, first note that the entries of infinite-matrix $Z$ are coefficients in an infinite series representation of a transition function, as seen by Equation 4.  This implies that the magnitude of entries of $Z$ decay as the row and column increase.  This naturally leads to our method of approximating the transition functions by truncating $Z$ in the same way that smooth periodic functions can be approximated to arbitrary accuracy by a truncated Fourier series.  Our paper focuses on accounting for the truncation error which arises from such an approximation and how it propagates to error in the final solution given by our algorithm.  This is necessary since a more accurate approximation of the transition dynamics (larger value of $k$) induces a higher statistical and computational cost.
>
> To answer your question of "how is k chosen", at a high-level we choose $k$ so that we have a sufficiently accurate approximation of the transition dynamics to correctly compute an optimal reward function.  As shown by Theorem 6, the value of $k$ therefore depends on the inherent properties of the MDP when represented by a fixed orthonormal basis.  Intuitively, if the transition functions vary rapidly and are similar to each other, then we will need a higher value of $k$ to give  a more accurate approximation in order to discern them.  You can see in Lemma 7, we give an example where we fix the orthonormal basis and show that under appropriate bounds on the partial derivatives of the transition functions, we can bound the truncation error of matrix $Z$ by $\epsilon$ when $k = O(1/\sqrt{\epsilon})$.  The value of $\epsilon$ needed for Algorithm 2 to work is given by Theorem 6.  We provide more intuition on the conditions of Theorem 6  starting from line 206.
>
> >"Lemma 5. Also here, the bounds on the \infty-norm of matrix \mathcal{F} as well as the \infty-norm of the difference between \mathcal{F} and \hat{\mathcal{F}}. How are they guaranteed to hold?"
>
> We show in our proof of Theorem 6 how the conditions of Lemma 5 can be satisfied using the conditions of Theorem 6 (which only depend on true MDP) by leveraging our previously given lemmas/theorems.  We recommend looking at the assumptions of Theorem 6 rather than the preceding intermediate lemmas to understand our overall assumptions.  Intuitively, we show in our proof of Theorem 6 how to combine our preceding results that 1) we can estimate the transition functions to arbitrary accuracy (Theorem 3), 2) we can get accurate approximations of the $F$ matrices from accurate $Z$ matrices (Lemma 4), and 3) accurate $F$ matrices along with some additional conditions allow us to recover an optimal reward function using linear programming (Lemma 5).  Theorem 6 combines these results into a unified statement which guarantees our method (Algorithm 2).
>
> >"In general, I found difficulties in getting convinced on the assumptions used in the presented results and quite abstract. I would greatly appreciate that the authors motivate/justify all the assumptions and show how/when they are satisfied in practice, based on the properties of the problems at hand. The same holds for Section 5. How are the conditions of Lemma 7 guaranteed? Is there a particular class of MDPs  in which it is satisfied?"
>
> Our paper provides a range of sufficient conditions for our guarantees, from general assumptions on the infinite-matrix representation to describing a specific class of MDPs where our results hold. Theorem 6 provides the most general conditions under which our algorithm is guaranteed to correctly solve the IRL problem with high probability.  These conditions do not specify the basis and only impose requirements on the infinite-matrix representation of the MDP.  We explain the intuition of these assumptions on lines 206-217.  In Section 5, we fix the basis to be the trigonometric basis and show how this allows us to guarantee the correctness of our method with a more concrete condition, i.e., an upper bound on the partial derivatives of the transition functions.  Finally, in Appendix C.1, we describe a simple class of MDPs which fulfill the conditions of Lemma 7, and we perform experiments using these MDPs.
>
> >"Theorem 8. The authors show that when considering multiple dimensions, the IRL problem, as presented, becomes tractable when the reward function decomposes linearly over the dimensions. I think this assumption is quite restrictive."
>
> Our method **does work** in the $d$-dimensional setting **without any additional assumptions**, as discussed on lines 257-260, with the caveat being that the computational and statistical complexity scales with $\exp(d)$.  However, on lines 261-264, we argue that this exponential scaling with the dimension is likely unavoidable in the general case.  We note that even for the statistically simpler task of Imitation Learning from Observations alone (ILFO), $\exp(d)$ samples and linear program constraints are needed (see section 5.1 of citation [1] provided by Reviewer aTgp). As we expand on in our response to Reviewer MtyF, determining the best simplifying assumptions to make the $d$-dimensional case polynomial in $d$ is likely domain specific and outside the scope of our inquiry.
>
> >"I would greatly appreciate it if the authors reorganize the core of the paper, focusing on clarity."
>
> We note that Reviewer aTgp remarked that our paper is "very well-written and easy to follow."  We would appreciate a more specific recommendation regarding what you found unclear, so we can address the point directly.
>
> << Minor Comments >> Thank you for pointing out the typos, we will fix them.

---

> > ### Comment · Reviewer_MeZM · 2021-08-22
> > **Re: Reply to initial review by Reviewer MeZM**
> >
> > I thank the authors for the detailed feedback. I think that my concerns have been addressed, thus I am raising my score. Regarding the reorganization of the paper, I believe that clarity would benefit from stating all the assumptions of Theorem 6 at the beginning of Section 3 and specify that the ones employed in the intermediate results are a consequence of those.

---

### Official Review · Reviewer_pRjN · 2021-07-15

**Rating:** 5
**Confidence:** 4

**Summary:**

This paper offers a theoretical perspective on Inverse Reinforcement Learning in the continuous (state) setting. The main contribution is to show that under some smoothness assumptions, they are able to recover a reward function with high probability.

**Limitations And Societal Impact:**

Yes.

**Main Review:**

This paper shows that in the continuous RL setting it is possible to recover a reward function with high probability in a single bounded dimensional state space, with discrete actions, by projecting the transition function (and an "equivalent" reward function on a functional basis). While the idea has some interest, its significance is limited for the following reasons.

- The setting is presented as continuous in the title of the paper, which is confusing. Usually, the continuous setting refers to a continuous state and action space rather than a continuous state space. For instance, the authors argue that this setting is of interest for robotics although it is precisely the continuous state action setting that would be of interest.

- A number of assumptions makes the approach impractical e.g. the estimateZ algorithm assumes that we can sample the state space uniformly, the expert policy is know in all states. Which reduces the interest of the paper in its theoretical contribution which itself is on a very contrived setup (one-dimensional state space).

- What is the point of discretizing the problem in Algorithm 2. step 4, where you could discretize it in the very first place (which is fine since transitions as smooth and the projected "equivalent reward" function is smooth), to transpose the problem to a discrete state space for which there are existing results.

- The section on adapting the approach to d-dimensional problem is not convincing. Assuming the independence of the transition function is not standard and as such should be verified by empirical evidence.

Nitpicks:
- You assume that we can swap actions to make the policy be constant. While this makes sense when there are no assumption on the structure of the mdp I don't think this is as natural if there are smoothness assumptions on the transitions. The authors could comment on that.
- r is a very poor choice of temporal subscript in the context of RL
- l414 -> "by linearity" is not an argument  in the context of a mathematical proof
- Even if mentioned later in the paper, in the proof of prop 1, exchanging an integral and an infinite sum is not trivial, you should mention why it is ok.

**Time Spent Reviewing:**

4h

---

> ### Author Response · Authors · 2021-08-11
> **Reply to initial review by Reviewer pRjN**
>
> Thank you for your detailed review.  We would like to clarify a few points to address your concerns.
>
> >"Usually, the continuous setting refers to a continuous state and action space rather than a continuous state space. For instance, the authors argue that this setting is of interest for robotics although it is precisely the continuous state action setting  that would be of interest."
>
> From our reading of the literature, it seems that "continuous setting" can refer to any combination of continuous time, state space, or action space.  Our abstract emphasizes that our contribution is in regard to a continuous state space and our definition of the MDP (Definition 1) defines the action space as discrete.  Hence, we argue that our focus is reasonably clear to any reader.
>
> We assert that our method for the non-parametric continuous state and discrete actions setting is a natural and necessary intermediate result towards provable methods for the continuous state-action setting.  As Reviewer aTgp recognizes, we "present the first algorithm for IRL in continuous setting with formal guarantees including sample and computational complexity bounds."
>
> Because of the lack of prior work, it is reasonable and precedented for us to focus on generalizing the state space at this point. The setting of a continuous state space and discrete action was explored previously for IRL by [1].  The development of theoretically validated methods for RL also supports our claim that our work is a natural intermediate step. The paper [2] by Brunskill et al. analyzed the sample complexity of a Probably Approximately Correct (PAC)-learning algorithm for reinforcement learning under unknown parametric transition dynamics in the continuous state and discrete action setting.  Brunskill et al. also motivated their work by potential applications to robotics. Follow-up work later provided a PAC-learning algorithm for RL that partially lifted the assumption of a discrete action space [3]. Building off of this work, a paper by Houthooft et al. presented a variational-based method for RL that achieved "significantly better performance compared to heuristic exploration methods across a variety of continuous control tasks and algorithms," [5]. We cite this line of research in RL as motivation for considering the continuous state discrete action setting, as ultimately we believe results for this setting can inform practical methods in domains such as robotics.  Our method for estimating the transition function can be immediately generalized to the case where the transition depends continuously on the action 'a'. Please see our reply to Reviewer MeZM starting with the phrase: "We believe that we could potentially extend our approach," for more information on this extension.
>
> >"A number of assumptions makes the approach impractical e.g. the estimateZ algorithm assumes that we can (1) sample the state space uniformly , (2) the expert policy is know in all states . Which reduces the interest of the paper in its theoretical contribution which itself is on a very contrived setup (3) (one-dimensional  state space)."
>
> 1.  The assumption that we can sample the state space uniformly has been used previously in both citation [10] in our paper and citation [3] provided by Reviewer MeZM.  This assumption relaxes the original assumption by [1], i.e., that the transition dynamics are known exactly.  In fact, **even in the discrete setting, we are unaware of any work which theoretically analyzes the sample complexity of an IRL algorithm under unknown non-parametric transition dynamics which does not use this assumption of uniform sampling.**
>
> 2.  The assumption of exactly knowing the expert policy is another standard assumption used in prior literature by both citation [10] and [13] in our paper.  Citation [3] provided by Reviewer MeZM assumes that the deterministic policy can be queried in any state.  Given that the total sample complexity is greater than the number of states, this is essentially the same as assuming the policy is known.  Therefore, **our assumption of knowing the expert policy is standard in theoretical analysis of IRL.**
>
> 3.  The statement that our results are limited to a one-dimensional state space is incorrect.  We show in Section 7 that **our approach is applicable in a d-dimensional state space** under a range of assumptions.
>
> Overall,  we argue that assumptions (1) and (2) are standard and allow us to focus on our core contribution of developing the first provable method for non-parametric IRL in the continuous state space setting using a previously unexplored infinite matrix formulation, which Reviewer aTgp described as "novel and inspiring" and Reviewer MtyF believes may "be useful for other theoreticians in the field".
>
> >"What is the point of discretizing the problem in Algorithm 2. step 4, where you could discretize it in the very first place ... for which there are existing results."
>
> To the best of our knowledge, there is no existing provably correct algorithm for IRL in our continuous setting, as Reviewer aTgp notes.  Therefore, we interpret your statement that there are "existing results" as referencing the few papers which provide similar guarantees in the discrete setting.  Please clarify if this interpretation is not correct.
>
> Extending guarantees from the discrete IRL setting to a continuous space by discretization has multiple drawbacks and is certainly non-trivial.  We argue that a truncation argument based on infinite series approximations is much more conceptually intuitive than a discretization approach.  Additionally, the extensive literature on function series approximations allows us to flexibly achieve stronger guarantees with stronger structure on the MDP class.  Due to space limitations, we refer to our response to Reviewer aTgp starting with the phrase: "While we believe it would be possible to apply," for a more comprehensive argument.
>
> >"Assuming the independence of the transition function is not standard and as such should be verified  by empirical evidence."
>
> We support this assumption on lines 267-272, where we show that this assumption was previously successful in applications of IRL to human navigation and is supported in the domain of human motor control.  Furthermore, Section 7 is meant to show that our work can be used to obtain strong guarantees on IRL in the d-dimensional setting, which we do by discussing a d-dimensional approach with no additional assumptions (lines 257-264) and the d-dimensional approach with a simplifying assumption for better complexity results.  We maintain that we successfully demonstrate the potential of our approach with these results, and that identifying the optimal assumptions for the d-dimensional setting is outside the scope of our work, since it is likely domain specific.  We will add clarification to our motivation for Section 7 in the text and some additional detail for the general d-dimensional method.
>
> >"You assume that we can swap actions to make the policy be constant. While this makes sense when there are no assumption on the structure of the mdp I don't think this is as natural if there are smoothness assumptions on the transitions."
>
> We note that the assumption was previously used for the continuous state discrete action setting by Ng and Russell and validated with a few simple experiments [1].  One interpretation of our assumption which avoid the action switching argument of [1] is that each transition function $P_a(s' | s)$ represents a composition of a policy at each state with the transition function so that $P_\pi(s' | s) = P(s' | s, a = \pi(s))$, and that this composition of the policy function with the transition function is smooth.  Then our method returns a reward function that makes a single policy strictly optimal compared to a finite set of alternative policies.  We agree that it would be interesting to see what extent this assumption can be relaxed or implemented.
>
> >"l414 -> "by linearity" is not an argument in the context of a mathematical proof"
>
> >"Even if mentioned later in the paper, in the proof of prop 1, exchanging an integral and an infinite sum is not trivial"
>
> We will address both of these concerns by being more explicit in our application of Fubini's theorem to justify interchanging infinite summations and integrals in our proof by applying results from section 3.7 of [5].  This will justify both mentioned steps.
>
> Line 414 will be rewritten to initially display the product of the infinite-matrix and infinite-vectors as an infinite summation.  Then, since each term is a product of a bounded term (the infinite-matrix/vector coefficient products) and the absolutely convergent infinite series $(\gamma^r)$, we can interchange the infinite summation in $r$ with the infinite summation representing the matrix-vector multiplication.  Rewriting this using our infinite-matrix/vector notation provides our current conclusion.
>
> For the interchange of the integral and infinite sum in prop 1, we will cite Fubini's theorem on a bounded interval (Theorem 3.7.12 from [5]).  We will point out that the infinite summation is equivalent to Lebegue integration with counting measure.  Then the interchange is justified, since the terms of the sum must be bounded by a constant, since the series is convergent, and the integral is taken over a bounded interval.
>
> [1] Ng, Andrew Y., and Stuart J. Russell. "Algorithms for inverse reinforcement learning." ICML. 2000.
>
> [2] Brunskill, Emma, et al. "Provably Efficient Learning with Typed Parametric Models." Journal of Machine Learning Research. 2009
>
> [3] Pazis, Jason, and Ronald Parr. "PAC optimal exploration in continuous space Markov decision processes." Twenty-Seventh AAAI Conference on Artificial Intelligence. 2013.
>
> [4] Houthooft, Rein, et al. "VIME: Variational Information Maximizing Exploration." NeurIPS. 2016
>
> [5] Benedetto, John J., and Wojciech Czaja. Integration and modern analysis. Springer, 2010.

---

> > ### Comment · Reviewer_pRjN · 2021-08-12
> > **Response to the authors**
> >
> > I thank the authors for their response. Under the specificity of the assumptions used, I have a hard time evaluating the significance of the work, and encourage the authors to demonstrate these assumptions are actually standard. More specifically:
> >
> > > that we can sample the state space uniformly has been used previously in both citation [10] in our paper and citation [3] >
> >
> > > The assumption of exactly knowing the expert policy is another standard assumption used in prior literature by both citation [10] and [13] in our paper >
> >
> > It is not exactly standard, since the authors provide 3 references, and [13] actually states that the more realistic setting is to assume that we have access to a set of expert trajectories. Another more realistic setting would be work such as DAGGER where the expert is queried under a some budget, on states visited while interacting with an environment. I would encourage the authors to point to significant work that operate under the same assumptions for the reviewers to grasp the significance of the setting.
> >
> > > We support this assumption on lines 267-272, where we show that this assumption was previously successful in applications of IRL to human navigation and is supported in the domain of human motor control. >
> >
> > If I understand correctly the references pointed out by the author, the assumption in these papers consists in representing a task into multiple factored tasks which is not exactly the same as assuming the independence of the transition function on all the dimensions of the state space. Again, a more extensive discussion on this assumption (if the authors do not deem necessary to introduce empirical evidence as explicitly suggested) would be appreciated.

---

> > > ### Author Response · Authors · 2021-08-19
> > > **Response to Reviewer pRjN**
> > >
> > > Thank you for your comments.
> > >
> > > > "It is not exactly standard, since the authors provide 3 references...Another more realistic setting would be work such as DAGGER where the expert is queried under a some budget"
> > >
> > > We emphasize that our paper is on solving the IRL problem, not on reinforcement learning or imitation learning.  That is, we are learning a reward function that makes an expert policy Bellman optimal, not the optimal policy.  As Reviewer MtyF notes, learning the reward function is a distinct problem that is interesting for its own purposes.  The reason we claim our assumptions are standard for theoretical analysis of IRL based on three papers ([1,2,3] below) is because those three papers make up the total body of prior work which studies the sample complexity and correctness of IRL under unknown non-parametric transition dynamics.  We point out that even the contemporaneous work published at ICML 2021 on analyzing IRL in the *discrete* setting uses assumptions equal to ours [3]. We argue that this is a more reasonable comparison on whether our work contributes to knowledge of IRL, rather than comparing our work to the settings used in entirely different problems.
> > >
> > > You point to the DAGGER method as an example of a more realistic setting, where the expert policy is queried under a budget.  However, we point out that the goal of the DAGGER paper is to learn an approximate policy $\widehat{\pi}$ with guarantees, so knowing the expert policy exactly would not make sense for this problem to begin with.  Meanwhile, our paper provides a way to learn a reward function (cost function in the DAGGER paper) that is Bellman optimal. Our method solves the IRL problem, which is interesting even when the policy is known exactly.  For example, our method can be used to provide a succinct and robust representation of a simulated agent's behavior as a continuous reward function, which is a useful but non-trivial task.  Neither the DAGGER method nor any other method we are aware of provides a theoretically validated solution to this problem.
> > >
> > > If one wants to provably recover a Bellman Optimal reward function in a setting where the expert policy is not known exactly, then our paper still provides a useful method with guarantees that can be extended to solve this problem.  For example, one can use the DAGGER method to query an expert and recover an exactly known policy $\widehat{\pi}$ that is an $\epsilon$-approximation to the unknown expert policy.  If this approximation to the expert policy is sufficiently accurate, then for purposes of our method, we can assume that we know the expert policy exactly.  Our main point is that the assumptions we use are standard in prior work on the problem we consider and such assumptions do not  diminish the impact of our work to any significant extent.  Our method has useful applications where the expert policy is known exactly (e.g. describing a simulated agent) and can be paired with other methods that approximate the expert policy (such as DAGGER) to recover a Bellman optimal reward function when the expert policy is unknown.
> > >
> > >
> > > > "If I understand correctly the references pointed out by the author, the assumption in these papers consists in representing a task into multiple factored tasks which is not exactly the same as assuming the independence of the transition function on all the dimensions of the state space. Again, a more extensive discussion on this assumption (if the authors do not deem necessary to introduce empirical evidence as explicitly suggested) would be appreciated."
> > >
> > > The experiments on human navigation through obstacles in the Modular IRL paper [4] decomposes the navigation tasks into 1) distance of the human to the nearest cylinder, 2) heading angle of the human, and 3) distance of the human from the center line of the walkway. Each of these variables is one-dimensional, and their experiments show that the decomposition of the total IRL problem into 1-dimensional IRL problems effectively produces a reward function which captures human navigation preference.
> > >
> > > The reason that we do not deem it necessary to empirically validate this assumption is that it is not essential to the core contributions of our paper.  We even discuss on Lines 257-264 than we can solve IRL problems in a $d$-dimensional setting without *any* additional assumptions on the transition dynamics.  Our paper provides a formalization of IRL in a non-parametric continuous setting which allows guaranteed recovery of a Bellman optimal reward function with computational and sample complexity results.  This formalization through infinite matrices can provide strong guarantees under many alternative assumptions and, as Reviewer MtyF notes, may be useful to other theoreticians in this field.  The point of Theorem 8 is to showcase the flexibility of our formalization and its potential to be used under different assumptions.  Empirically validating the assumption of Theorem 8 does little to support our claim that our formalization is useful for further theoretical research on IRL, since whether this particular assumption is empirically validated or not does not change whether other theoreticians can use our formalization to prove guarantees for IRL under other assumptions. However, given that several reviewers have asked questions regarding Theorem 8, we would like to propose that we provide a more sophisticated extension to the $d$-dimensional setting, thereby more clearly exhibiting our theoretical contribution.
> > >
> > > Our new extension would guarantee the correctness of our method in the case where the $d$-dimensional transition functions of the given MDP can be decomposed into the product of $T$ factor transition functions where the $i$-th factor operates on a subspace of the total state space, $S^{(i)} \subset S$, such that $\dim(S^{(i)}) \leq q$ and $S = S^{(1)} \times S^{(2)} \times ... \times S^{(T)}$. In other terms, $P(s_1 | s_0) = \prod_{i=1}^T P_i(s_1^{(i)} | s_0^{(i)})$ and each component transition function in the product is over a state space of at most dimension $q$. Using our current results with a few slight modifications, we can guarantee a solution to this problem with $O(T \exp(q))$ scaling to our current complexity results.
> > >
> > > This formulation encompasses both approaches we currently discuss in Section 7.  Notice that if we use one factor of $d$-dimensions ($T = 1$ and $q = d$), then we get the $\exp(d)$ scaling method that makes no additional assumptions on the transition functions, i.e., the original $d$-dimensional problem.  If we decompose the transition function as a product of $d$ one-dimensional transition functions ($T = d$ and $q = 1$), then we recover the same conclusion as Theorem 8, i.e., we can solve the total IRL problem by solving $d$ one-dimensional IRL problems.  We briefly describe what we would change for this conclusion.
> > >
> > > We first expand on Lines 257-264 of our paper (the T=1, q=d case) and explain in more detail how one can solve a $d$-dimensional IRL problem without any additional assumptions. To start with, we would change our orthonormal basis functions to be over a $d$-dimensional space such that $\phi_i(s):[-1, 1]^d \rightarrow [-1, 1]$.  This can be done by letting each $\phi_i$ to be the product of $d$ one-dimensional basis functions.  The key insight is that our proofs rely on representing $\int_{s_1\in S} P(s_2 |s_1) P(s_1|s_0) ds_1$ as a product of infinite matrices and the expectation $\int_{s_1 \in S} R(s_1) P(s_1|s_0) ds_1$ as an infinite matrix-vector product.  Since this holds equally well when $S$ is $d$-dimensional, we don't need to change any of our results prior to Lemma 5.  In Lemma 5, we use a $c$-cover, $\bar{S}$, of $[-1, 1]$ and $\rho$-Lipschitz continuity of $\widehat{\alpha} \widehat{\mathcal{F}} \phi(s)$ to lower bound the Bellman optimality of our returned reward function.  This same argument holds by using a $c$-cover of the $d$-dimensional state space and $\rho$-Lipschitz continuity of $\widehat{\alpha}^\top \widehat{\mathcal{F}} \phi(s)$ with respect to the $\ell_2$-norm. Guaranteeing that $\alpha^\top \mathcal{F} \phi(s)$ is Lipschitz in the $d$-dimensional setting is no more difficult than guaranteeing it in the one-dimensional setting, since guaranteeing that it is $\frac{\rho}{\sqrt{d}}$-Lipschitz in each dimension guarantees it is $\rho$-Lipschitz with respect to the $\ell_2$-norm.  This part on creating a $c$-covering for the $d$-dimensional state space is where the $\exp(d)$ computational scaling will occur, since the covering number of a $d$-dimensional hypercube is $\exp(d)$, we will need $\exp(d)$ points in our cover and hence $\exp(d)$ constraints in the linear program we solve. However, the proof remains the same.
> > >
> > > The above modifications shows that our method can be used to solve a $d$-dimensional IRL problem in $\exp(d)$ time without additional assumptions.  To extend our method to general values of $T$ and $q$, notice that our proof of Theorem 8 only assumes that the transition functions of an MDP can be decomposed, not that component transition functions are one-dimensional.  Therefore, we can immediately conclude by the proof of Theorem 8 that we can solve a $d$-dimensional IRL problem by solving each of the $T$ IRL problems in the decomposition in $\exp(q)$ time.
> > >
> > > [1] Ng, Andrew Y., and Stuart J. Russell. "Algorithms for inverse reinforcement learning." ICML. 2000
> > >
> > > [2] Abi Komanduru and Jean Honorio. On the correctness and sample complexity of inverse
> > > 325 reinforcement learning. In NeurIPS, 2019.
> > >
> > > [3] Metelli, Alberto Maria, Giorgia Ramponi, Alessandro Concetti, and Marcello Restelli. "Provably Efficient Learning of Transferable Rewards." ICML. 2021.
> > >
> > > [4] Constantin A Rothkopf and Dana H Ballard. Modular inverse reinforcement learning for
> > > 334 visuomotor behavior. Biological cybernetics, 2013.

---

### Official Review · Reviewer_MtyF · 2021-07-21

**Rating:** 6
**Confidence:** 2

**Summary:**

The paper focuses on the theory of inverse reinforcement learning and provides a solution for a bounded continuous space MDP with unknown transition dynamics while maintaining formal guarantees. To this end, the paper proposes to use an infinite sum of orthogonal basis functions to represent the estimated dynamics which allows using a linear program to recover the reward functions. Instead of the standard functional analysis approach, the paper uses the toolset of infinite matrices claiming that it allows them to derive more meaningful error bounds. More practically, they demonstrate that if the transition functions can be represented by truncated matrix of coefficients, a solution to the IRL problem can still be recovered with bounded probability. The derived sample complexity in the continuous state space matches that of the earlier proven discrete case. The paper also provides an efficient extension for $d$-dimensional setting under some assumptions. The paper also performs some simple experiment to evaluate the bounds on randomly generated IRL problems.

**Limitations And Societal Impact:**

It's a theory paper and it adequately mentions its limitations.

**Main Review:**

Overall, it's a hard paper for me to judge as I am no expert in the mathematical toolsets used in this paper.

**Strengths**

IRL is a difficult problem in general, and [1,2] showed ways to avoid the problem. But for a lot of tasks we do care about the reward function and not just the policy. This paper is amongst the few that try to carve out a space of MDPs where this reward inference problem can be solved with some guarantees. The paper uses very different math to achieve this which might be useful for other theoreticians in the field to explore.

**Weaknesses**

Major portion of the results hinges on the specific orthonormal basis function that can be used for representing the transition dynamics and they still don't really work for the $d$ dimensional case except under strong separability of the dimensions which doesn't sound realistic for most problems. Source code would have been immensely useful to better understand how the practical inference of transition dynamics works using this theory to allow setting up the linear program for reward functions. It's hard for me to judge whether results in Fig 2 suggest that the proven bounds are fairly tight or not.

[1]: J Ho and S Ermon. "Generative Adversarial Imitation Learning" https://arxiv.org/abs/1606.03476

[2]: J Ho et. al. "Model-Free Imitation Learning with Policy Optimization" https://arxiv.org/abs/1605.08478

**Time Spent Reviewing:**

3

---

> ### Author Response · Authors · 2021-08-11
> **Reply to initial review by Reviewer MtyF**
>
> Thank you for your comments on our paper.  We appreciate your acknowledgment of the novelty of our approach, and we agree that recovering an optimal reward function is an important, yet theoretically understudied task.  We agree that the infinite-matrix formulation we develop may be useful for further development of theory in this area.  We would like to clarify a few points to address your concerns.
>
> > “Major portion of the results hinges on the specific orthonormal basis function”.
>
> We interpret this point as regarding the need to specify an orthonormal basis so that the transition dynamics of the MDP can be efficiently represented, since we do not require the use of any particular orthonormal basis.
>
> We view the dependence of our work on using an appropriate orthonormal basis as a natural and necessary approach to leverage structure of specific MDP classes, such as sparsity or decay of transition function coefficients over a given basis.  To better address your point, we would like to ask what alternative approach do you have in mind that could be used to provably recover an optimal non-parametric reward function?  Without using a specific basis, it is unclear how one can efficiently represent a smooth non-parametric reward function.  While Reproducing Kernel Hilbert Space (RKHS) approaches have been used for non-parametric methods in other tasks, the task of IRL is more complicated than typical applications of RKHSs we are aware of, since we must return a function.  Additionally, one known drawback of RKHS approaches is that their complexity tends to scale with the amount of data/number of samples.  We believe that inorder to make such an approach work, one would have to impose smoothness and function complexity assumptions that would essentially be equivalent to our assumptions.  Since orthonormal bases have been extensively studied and shown to provide efficient representations of objects in other tasks, such as the wavelet basis in image compression, we believe that being explicit about using an orthonormal basis is advantageous and allows strong finite sample guarantees.
>
> > “they still don't really work for the d dimensional case except under strong separability of the dimensions”
>
> Our method **does work** in the d-dimensional setting **without any additional assumptions**, as discussed on lines 257-260, with the caveat being that the computational and statistical complexity scales with exp(d).  However, on lines 261-264, we argue that this exponential scaling with the dimension is likely unavoidable in the general case.  We note that even for the statistically simpler task of Imitation Learning from Observations alone (ILFO), exp(d) samples and linear program constraints are needed (see section 5.1 of citation [1] provided by Reviewer aTgp).
>
> The main point of Section 7 is to demonstrate that our results can be translated to finite sample guarantees in d-dimensions, in generality or with simplifying assumptions for better complexity results.  We motivate our example simplifying assumption on line 267-272 by previous work which shows it is empirically supported regarding human motor control (citation [8] in our paper) and that the assumption was previously used in an IRL method to predict human navigation preference (citation [14] in our paper).  However, focusing on determining the optimal set of assumptions for the d-dimensional case is outside the scope of our paper, since it would likely be domain specific.  We will add more clarification on our motivation for Section 7 and the details of the exp(d) scaling d-dimensional method, as we have just described.
>
> > “Source code  would have been immensely useful to better understand how the practical inference of transition dynamics works using this theory to allow setting up the linear program for reward functions. It's hard for me to judge whether results in Fig 2 suggest that the proven bounds are fairly tight or not.”
>
> We will be able to fully release our source code with documentation before the camera-ready deadline.  Additionally, we provide the source code we use to compute the empirical Z-hat matrices we use to set up our linear program for the reward function that you describe.  As you can see below, the code needed to implement our estimateZ algorithm is minimal. We note that our concentration analysis for the entries of Z-hat use Hoeffding’s inequality, which only requires that a random variable has support on a bounded interval, hence we can expect the entries in our case to concentrate faster than our worst case bounds imply.  Although, we do not think a tighter analysis of this concentration would change our complexity results.
> ```
> def sample_Z(model, size, N):
>
>     Zhmat = np.zeros((2*size + 1, 2*size + 1))
>
>     for j in range(N):
>
>         start_s = np.random.uniform(-1, 1)
>         next_s = model.sample(start_s)
>
>         Zhmat += (2/N) * np.outer(phi_vec(next_s, size), phi_vec(start_s, size))
>
>     Zhmat[0,0] = 1/2    # Constant function coefficient
>
>     return Zhmat
> ```

---

> > ### Comment · Reviewer_MtyF · 2021-08-19
> > **Thanks for the response**
> >
> > Thank you for clarifying d-dimensional case. Although I am still not sure if the statistical complexity _has_ to scale with $exp(d)$ or it's just because of your framing of the problem. I guess you also are only claiming _likely_, so more work is likely required. Thanks for the Z sampling code, very useful!
> >
> > Your interpretation of my comment was correct. And I see your point that it's unlikely that we can avoid this assumption for a theoretical analysis.

---

### Decision · Program_Chairs · 2021-09-27

**Decision:**

Accept (Poster)

**Comment:**

This paper provides an IRL algorithm for control processes with continuous states, discrete controls, and unknown transition dynamics with formal sample/time complexity guarantees. The authors addressed many of the initial concerns raised by the reviewers. The primary remaining ones are: (1) clarity and organization of the work---particularly where the assumptions are introduced; (2) the standardness and practicality of the assumption of a known and strictly optimal expert policy (rather than trajectory samples). The first issue seems possible to resolve in the paper's revision. For the second issue, the reviewers in discussion tended to view the paper as making an important theoretical contribution that could lead to extensions with more realistic/relaxed assumptions or serve as a bridge toward continuous state and control settings. Thus, there is still significant merit to the paper even with these concerns remaining.

Given all of this, I recommend (weak) acceptance for the paper, but consider it to be the most borderline of the papers I am recommending for acceptance and I am not opposed to papers with stronger proponents being prioritized above it as space limits may require.

As a minor comment to the authors: I agree with Reviewer pRjN that the paper title could potentially be misleading and suggest revision to "... Continuous State ..."